# Mechanical stress determines the configuration of TGFβ activation in articular cartilage

Gehua Zhen[1], Qiaoyue Guo[1], Yusheng Li[1], Chuanlong Wu[1], Shouan Zhu[1], Ruomei Wang[1], X. Edward Guo[2], Byoung Choul Kim[3], Jessie Huang[4], Yizhong Hu[2], Yang Dan[1], Mei Wan[1], Taekjip Ha[3], Steven An[4,5] & Xu Cao[1✉]

Our incomplete understanding of osteoarthritis (OA) pathogenesis has significantly hindered the development of disease-modifying therapy. The functional relationship between subchondral bone (SB) and articular cartilage (AC) is unclear. Here, we found that the changes of SB architecture altered the distribution of mechanical stress on AC. Importantly, the latter is well aligned with the pattern of transforming growth factor beta (TGFβ) activity in AC, which is essential in the regulation of AC homeostasis. Specifically, TGFβ activity is concentrated in the areas of AC with high mechanical stress. A high level of TGFβ disrupts the cartilage homeostasis and impairs the metabolic activity of chondrocytes. Mechanical stress stimulates talin-centered cytoskeletal reorganization and the consequent increase of cell contractile forces and cell stiffness of chondrocytes, which triggers αV integrin–mediated TGFβ activation. Knockout of αV integrin in chondrocytes reversed the alteration of TGFβ activation and subsequent metabolic abnormalities in AC and attenuated cartilage degeneration in an OA mouse model. Thus, SB structure determines the patterns of mechanical stress and the configuration of TGFβ activation in AC, which subsequently regulates chondrocyte metabolism and AC homeostasis.

[1] Department of Orthopaedic Surgery, The Johns Hopkins University, Baltimore, MD, USA. [2] Department of Biomedical Engineering, Columbia University, New York, NY, USA. [3] Department of Biophysics and Biophysical Chemistry, The Johns Hopkins University, Baltimore, MD, USA. [4] Department of Pharmacology, Rutgers-Robert Wood Johnson Medical School, The State University of New Jersey, Piscataway, NJ, USA. [5] Rutgers Institute for Translational Medicine and Science, New Brunswick, NJ, USA. ✉email: xcao11@jhmi.edu

Osteoarthritis (OA) is the most common degenerative joint disease and the leading cause of physical disability. Although the estimated prevalence of OA varies across studies, there is agreement that substantial proportions of adults are affected[1,2]. A recent report showed that the global age-standardized prevalence of knee OA and hip OA were 3.8 and 0.85%, respectively[3]. Because of a lack of disease-modifying therapy, patients with OA typically undergo joint replacement surgery at the end stage of the disease after experiencing joint pain and stiffness for decades[4,5]. Enormous effort has been devoted to studying OA pathogenesis, and many intrinsic signaling pathways of articular cartilage (AC) degeneration have been identified[6,7]. However, targeting the pathways has not generated any therapy that halts the degeneration of AC. As a skeletal tissue, AC is constantly challenged by mechanical stress. While AC buffers the mechanical force from the diarthrodial joint, it also receives the counter-force that is transmitted by the underlying subchondral bone (SB)[8]. Aberrant mechanical load is one of the primary etiological factors that leads to AC degeneration[9,10]. The biomechanical interplay between SB and AC and the molecular and cellular mechanisms that underlie the transduction of physical signals to biochemical signals remains to be elucidated.

In our previous study, we found that uncoupled remodeling resulted in excessive transforming growth factor beta (TGFβ) activation, which induced the clustering of osteoprogenitors and angiogenesis in the bone marrow cavity and abnormal formation of SB at the onset of OA[11]. Importantly, SB abnormality is highly correlated with cartilage degeneration in both OA animal models and humans with OA[11]. AC degeneration in OA animal models was ameliorated when the microarchitecture of SB was improved, suggesting that normal SB structure is essential for the homeostasis of AC[12,13]. We found that the incremental increase of either SB size or subchondral plate stiffness modulus results in a significant elevation of peak stress on the articular surface. However, we still do not understand how the mechanical signals in AC that are generated during the interplay with SB are transformed into biological signaling for AC homeostasis or degeneration in OA.

To accommodate the environment of constant mechanical challenge, chondrocytes monitor and regulate their metabolic activity in response to changes in the extracellular mechanical environment[14,15]. TGFβ is critical for the anabolic activity of chondrocytes. Interestingly, inhibition of TGFβ activity abolishes the anabolic activity of mechanical loading[16,17], suggesting that TGFβ may also play an essential role in propagating the mechanical signals in AC. TGFβ has been shown to promote chondrocyte proliferation and extracellular matrix (ECM) protein synthesis and release[18–22]. Genetic inactivation of TGFβ type-II receptor or its downstream factor, Smad3, leads to early onset of OA[23–25]. Conversely, intra-articular injection of recombinant TGFβ1 into mouse knee joints accelerated the development of OA[26]. Constitutive overexpression of active TGFβ1 resulted in hyperplasia of synovium and osteophyte formation[27]. Moreover, TGFβ was reported to inhibit the insulin-like growth factor-1 (IGF-1) signaling pathway while IGF-1 is a well-accepted anabolic factor that promotes matrix protein synthesis in chondrocytes[28–31]. TGFβ also induces the redox imbalance by increasing mitochondrial reactive oxygen species (ROS) production in various cell populations, including chondrocytes, which results in cell apoptosis, senescence, fibrotic gene expression, and more[32–35]. Mechanical loading compromises the chondrocyte respiratory function and results in decreased adenosine triphosphate (ATP) levels, as well as proton leakage and ROS formation[36]. Indeed, superposition of mechanical compression impairs the anabolic effect of TGFβ on chondrocyte matrix

production[37,38]. These observations indicate that the maintenance of proper levels of TGFβ is essential for AC homeostasis, whereas excessive or inadequate activation of TGFβ in AC are both detrimental. Interestingly, the nonlinear effect of TGFβ also has been observed in bone and other tissues[39,40].

TGFβ is deposited in AC matrix upon secretion by chondrocytes as an inactive form, and downstream signaling pathways are triggered when the active form of TGFβ binds to its receptor on the chondrocyte membrane[41]. The unique structure of latent TGFβ complex enables it to change its biological activity in response to mechanical stimuli. The latency-associated peptide (LAP) is the essential component within the latent TGFβ complex that masks the activity of TGFβ. A conformational change of LAP can result in the liberation of active TGFβ from the sequester of LAP[42]. The exact mechanism for TGFβ activation in chondrocytes, particularly in an active mechanical environment, has not been studied. It has been shown that the arginylglycylaspartic acid (RGD) motif in LAP is the natural ligand for the αV subunit containing integrins[43,44]. Cell contractile forces may induce a conformational change of LAP and consequent TGFβ activation through the bond between αV integrins and the RGD sequence in LAP[45,46]. A recent study showed that the required minimal force for inducing the conformational change of LAP is approximately 40 pN[47]. Therefore, to physically activate TGFβ, a cell should be able to exert contractile forces >40 pN, and the bond between αV integrin and RGD should be able to withstand this minimal required force. In eukaryotes, the cytoskeletal system is responsible for maintaining cell morphology and exerting forces through polymerized actin filaments[48]. Talin-centered focal complex connects the intracellular domain of integrins and actin filaments[49,50]. Moreover, the binding of talin to the cytoplasmic domain of integrin is the key step for integrin clustering and activation[51,52].

In our present study, we investigated how SB alterations affect AC homeostasis, focusing on the molecular signaling pathway that propagates the mechanical force in AC, particularly during OA development. We found that aberrant SB architecture leads to a redistribution of mechanical stress on AC. The pattern of mechanical stress is closely correlated with the pattern of TGFβ activity in cartilage. Mechanical stress significantly contributed to αV integrin-mediated TGFβ activation in chondrocytes. Mechanical force is responsible for the strengthening of the talin-centered cytoskeleton reorganization and consequent αV integrin activation. Knockout of αV integrin reversed abnormal TGFβ activation and attenuated abnormal mechanical loading-induced cartilage degeneration in OA.

## Results

**Increase of TGFβ activation in AC is correlated with the alteration of SB microarchitecture.** To examine the potential relationship between SB microarchitecture and TGFβ activity in AC, we dissected AC and the underlying SB from the knee joints of healthy donors and patients with OA undergoing knee replacement. In each specimen, we categorized AC degeneration as severe (OA-S), minimal (OA-M), or absent (healthy control [HC]). Immuno-staining of pSmad2/3 revealed that the TGFβ activity was unevenly distributed in osteoarthritic AC. The ratio of pSmad2/3-positive cells to total chondrocytes was significantly higher in the OA-S group than that in the OA-M or HC group (Fig. 1a, b). In parallel, micro-computed tomography (μCT) analysis showed that SB microarchitecture was most altered in the OA-S group, as indicated by significantly increased bone volume fraction (trabecular bone volume per tissue volume (BV/TV)) and decreased structural model index and bone surface/volume fraction (Fig. 1a, c–e). We also examined the association between

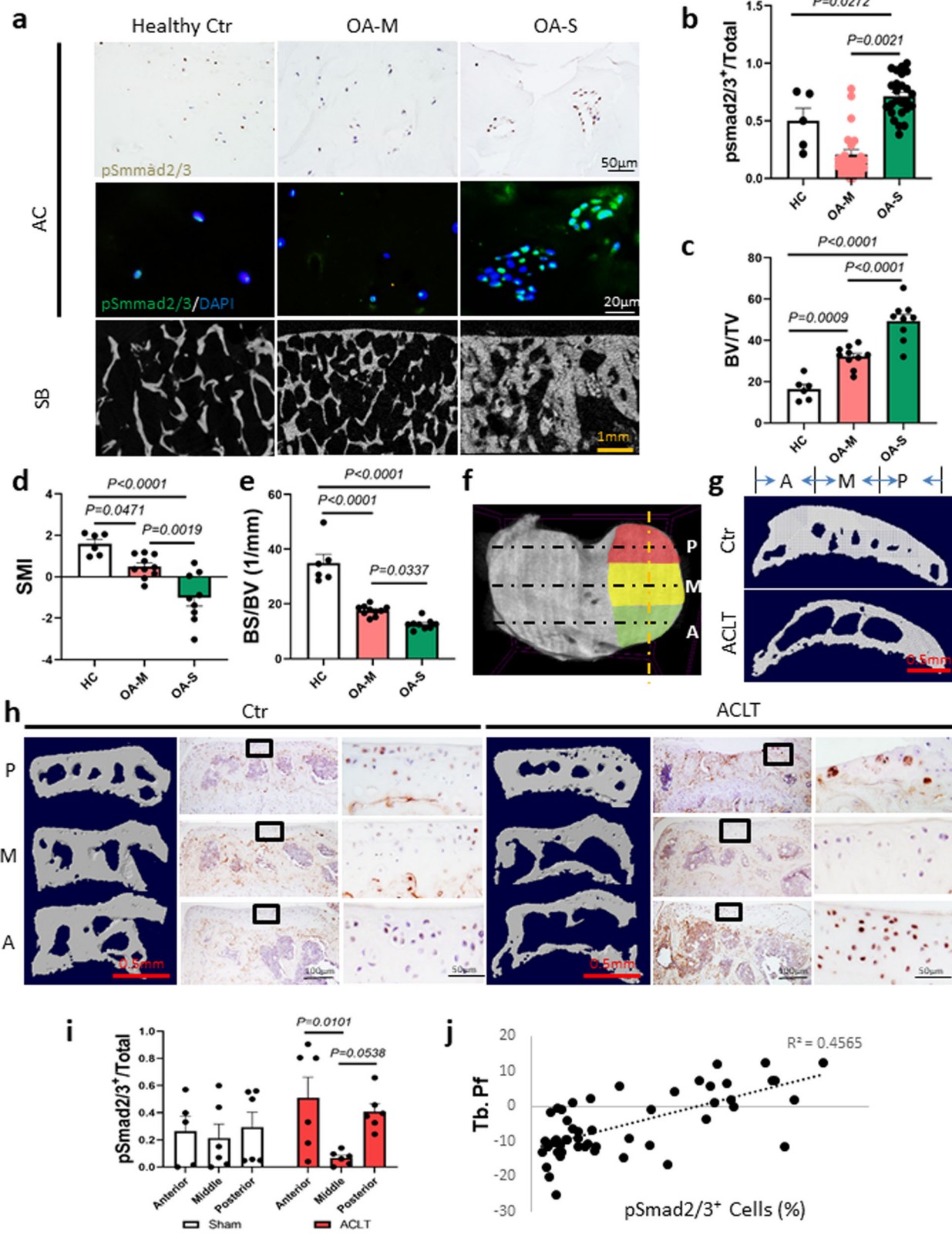

TGFβ activity in AC and the underlying SB structural abnormality in the mouse anterior cruciate ligament transection (ACL-T) OA model. The medial tibia plateaus of the mice were divided into three regions (anterior, middle, and posterior), and the structural indices were analyzed independently (Fig. 1f, g). The structure between OA and sham-operated mice was significantly different in the anterior region, whereas the differences in the middle and posterior regions were limited (Fig. 1g, h). Histological analysis showed that the anterior region showed the earliest signs of cartilage degeneration (Supplementary Fig. 1). Similar to human OA specimens, in OA mice the distribution of TGFβ activity in AC was altered significantly, whereas it was relatively

**Fig. 1 Excessive activation of TGFβ in AC was correlated with the structural alteration of SB in OA. a** Immunohistological staining (first row, pSmad2/3: brown), immunofluorescence staining (second row, pSmad2/3: green, DAPI: blue) staining of pSmad2/3 in AC, and representative μCT images of SB (third row) of human knee joint specimens. HC healthy control, OA-M OA specimen with minimal cartilage degeneration, OA-S OA specimen with severe cartilage degeneration. **b** Quantitative analysis of the ratio of pSmad2/3$^+$ cells in total chondrocytes in AC, $n = 5$ or 25 biologically independent joint samples in HC or OA-M and OA-S group, respectively. **c–e** Quantitative analysis of μCT scanning: BV/TV: percent bone volume, BS/BV: bone surface/ volume ratio, SMI: structure model index, $n = 6$, 10, and 9 biologically independent joint samples in HC, OA-M or OA-S group, respectively. In (**b–e**), data were analyzed using one-way ANOVA followed by LSD post hoc test. Data are presented as mean values $+/-$ SEM. Source data are provided as a Source data file. **f, g** The region-defining strategy for medial compartment of mouse tibia. **f** Top view of tibia plateau for illustration. The gold dotted line represents the location of the sagittal images in **g**, the black dotted lines represent the locations of the coronal images in **h**. P posterior, M middle, A anterior. **h** Representative 3D reconstructed μCT images of coronal sections of the medial compartment of mice tibia (first and fourth columns), and immunohistological staining of pSmad2/3 (brown) in AC at the corresponding locations (second, third, fifth, and sixth columns). The third and sixth columns are the high-magnification images of the box region in columns 2 and 5, respectively. The mice were sacrificed 2 months after sham surgery or ACL-T. **i** The ratio of pSmad2/3$^+$ cells to total chondrocytes in different regions of AC in **h**. Data were analyzed using two-way ANOVA Tukey's post hoc test. **j** The correlation of trabecular pattern factor (Tb. Pf.) of the SB and the percentage of pSmad2/3$^+$ cells in the above cartilage at the same location. **h–j** $n = 6$ biologically independent animals, data are presented as mean values $+/-$ SEM.

even in sham-operated mice. Specifically, the number of pSmad2/3$^+$ cells increased significantly in the anterior region and was moderately elevated in the posterior region but remained low in the middle region (Fig. 1i). Using trabecular pattern factor (Tb. Pf.) as an index for SB microstructure, we found a linear correlation between Tb. Pf. and the number of pSmad2/3$^+$ cells in the AC (Fig. 1j). These results suggest that structural changes in osteoarthritic SB are highly correlated with the pattern of TGFβ activation in AC.

**TGFβ activation pattern in AC is aligned with mechanical stress distribution and influences chondrocyte metabolic activity.** To understand how structural changes in SB regulate TGFβ activity in AC, we performed computerized finite element analysis (FEA) to map the mechanical stress distribution in AC. The stress levels and distributions in AC that transmitted from SB were simulated in the FE model based on the microstructure of SB. The three-dimensional (3D) reconstructed μCT images showed that the microstructure of SB in OA mice was distinct from that of the sham-operated mice. Compared with the sham-operated mice, the overall SB structure was altered at the interface between SB and AC, with a rugged appearance in OA mice (Fig. 2a). The SB trabeculae were more porous with the disorganized structure in OA mice relative to the sham group (Fig. 2a and Supplementary Movie 1). In conjunction with observations of a more plate-like structure, the SB displays a sclerotic bone phenotype with uneven distribution of bone mass and local stiffening. As a result, the simulated maximum principle strain and von Mises stress in OA mice were markedly elevated in the anterior and interior regions of the medial plateau, whereas they were slightly decreased in the center region (Fig. 2b, c). To further examine the relationship between the distribution of mechanical stress and TGFβ activity in AC, we analyzed TGFβ activity from the immunofluorescence intensity of pSmad2/3 in the whole layer of AC under a confocal microscope and correlated it with levels of mechanical stress at the corresponding area based on the FEA simulation. The pattern of TGFβ activity was highly correlated with the distribution of mechanical stress in AC (Fig. 2c, d). The fluorescent intensity of pSmad2/3$^+$ cells was elevated significantly in the anterior and interior regions, with higher mechanical stress relative to that in the center and posterior regions (Fig. 2c, d). The homeostasis of AC is a dynamic balance between cartilage anabolism and catabolism. To investigate the effect of excessive activation of TGFβ under mechanical stress on cartilage metabolism, we examined the metabolic activities in chondrocytes cultured in low (2 ng/ml) or high (10 ng/ml) concentration of active TGFβ1. We found that glucose uptake increased with the treatment of active TGFβ1 (Fig. 2e).

However, ATP generation did not match the increase of glucose uptake in chondrocytes treated with high concentrations of TGFβ1. Compared with the control group, ATP production was higher in chondrocytes cultured in low-concentration TGFβ and lower in chondrocytes cultured in high-concentration TGFβ (Fig. 2f). We then measured the intracellular pH as an indicator of the levels of glycolysis because lactic acid, the by-product of glycolysis, is known to decrease intracellular pH. pHrodo fluorescence staining revealed that intracellular pH in the high-concentration TGFβ group was significantly decreased in the majority of the cells and this phenomenon was not observed in the controls and the low-concentration TGFβ group (Fig. 2g). Leaking of free radicals from the respiratory chain often occurs simultaneously with a decrease in ATP production in mitochondria, so we used dihydroethidium (DHE) as a probe to examine the ROS production. We found that a high concentration of TGFβ increased ROS production significantly relative to the control group (Fig. 2h). Together, our findings indicate that the alteration of mechanical stress distribution in AC that is induced by the compromised structure in SB was closely associated with TGFβ activity in AC. Moreover, high levels of TGFβ1 significantly alter the metabolic activity of chondrocytes, particularly in an environment with mechanical stress.

**αV integrins mediate mechanical stress-induced activation of TGFβ in AC.** We then investigated how mechanical stress induces activation of TGFβ in AC. We found that the upregulation of TGFβ signaling in OA-S cartilage was correlated with the positive expression of αVβ6 integrins by double immunofluorescence staining of αV integrins and pSmad2/3 in AC from healthy donors and OA patients (Fig. 3a, b). Approximately 63% of the pSmad2/3$^+$ cells were also positive for αVβ6 in OA-S cartilage, whereas the αVβ6 and pSmad2/3 double-positive cells were rarely observed in the OA-M and HC groups (Fig. 3a, b). Similar findings were observed in OA animal models. The number of pSmad2/3$^+$ cells was almost doubled in the AC of OA mice relative to sham-operated mice, with a significant increase of αVβ6$^+$ cells (Fig. 3c–e and Supplementary Fig. 2). Notably, the pSmad2/3 and αVβ6 double-positive cells accounted for approximately 50% of the pSmad2/3$^+$ cells, a percentage similar to the increase of pSmad2/3$^+$ cells in total chondrocytes (Fig. 3c, d, f), suggesting a possible role of αVβ6 integrin in increased TGFβ activity in osteoarthritic cartilage. Immunohistological staining of other αV-containing integrins showed that chondrocytes that were positive for αVβ3 and αVβ5 integrins were also increased in AC of OA mice relative to the sham controls (Supplementary Fig. 3), suggesting that αVβ6 might not be the

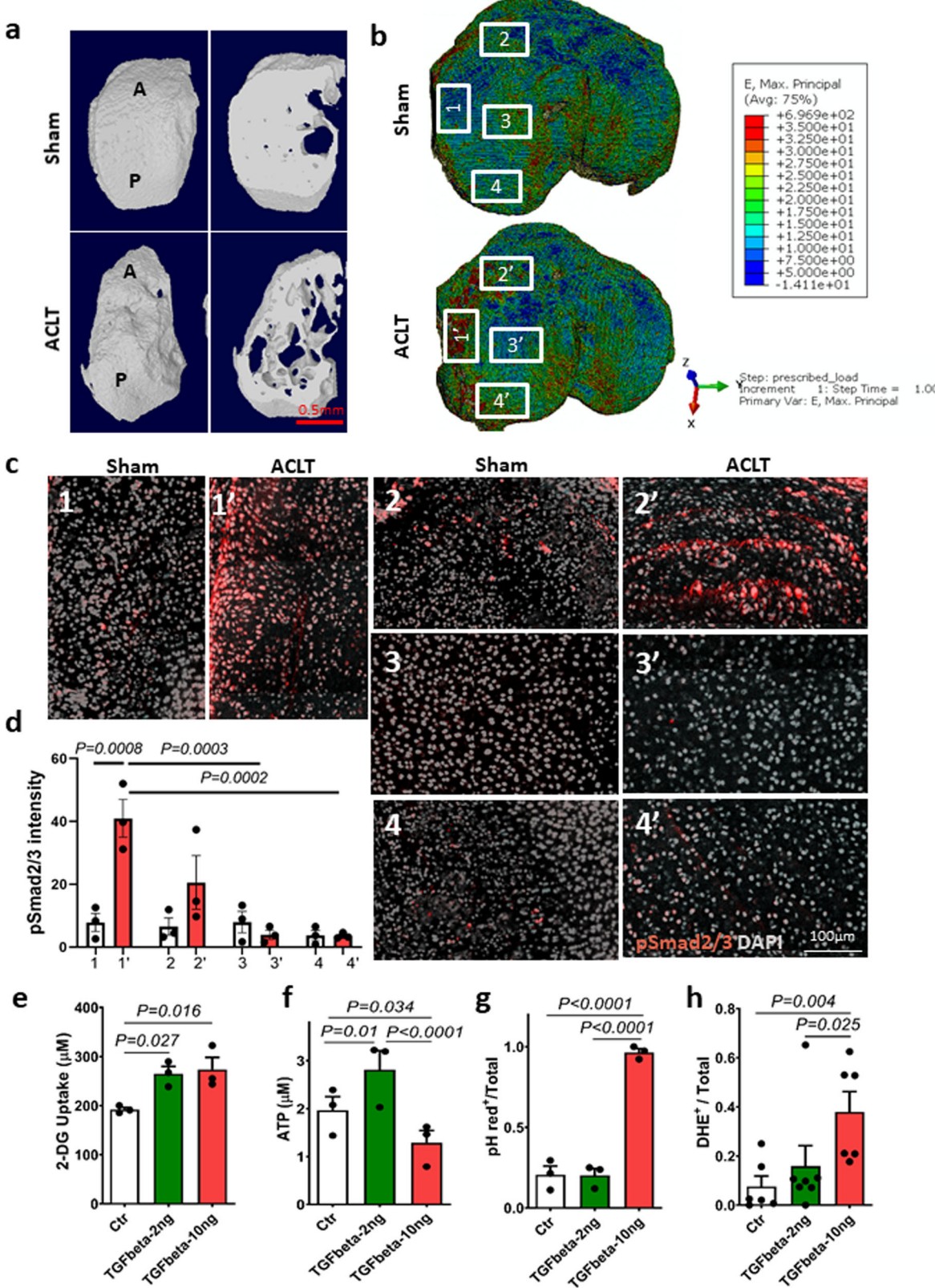

only αV-containing integrin that is involved in TGFβ activation in osteoarthritic AC.

We then examined whether αV integrin plays a critical role in mechanical stress-induced activation of TGFβ. We found that the expression of αV integrin was significantly increased when chondrocytes (Supplementary Fig. 4) were subjected to

continuous shear stress for 48 h. Moreover, the levels of pSmad2/3 in chondrocytes also increased in the mechanical stress+ group (Fig. 3g). Importantly, knocking down the expression of αV integrin using small interfering ribonucleic acid (siRNA) abolished mechanical stress-induced activation of TGFβ in western blot analysis (Fig. 3g). The levels of active TGFβ in the

**Fig. 2 TGFβ activation pattern in AC was aligned with mechanical stress distribution. a** Representative 3D reconstructed images of the tibia medial compartment from mice 1 month after ACL-T or sham surgery. The left images in each panel: top view; right images in each panel: transaxial view of the tibial plateau sectioned at the same horizontal level. **b** FEA simulation of the maximum principle strain at the tibia plateau based on the SB structure. The numbered boxes represent the individual areas in **c**. Source data are provided as a Source data file. **c** Immunofluorescence images of pSmad2/3 staining in full layer of AC of mouse tibia medial plateau at regions with different mechanical stress. The location of the image was indicated in **b**, pSmad2/3 (red), DAPI (gray). **d** Quantitative analysis of pSmad2/3 fluorescent intensity of **c**. $n = 3$ biologically independent animals, data were analyzed using two-way ANOVA Tukey's post hoc test. Data are presented as mean values $+/-$ SEM. **e–h** SV40 chondrocytes were cultured in low (2 ng/ml) or high (10 ng/ml) concentration of recombinant TGFβ1 for 48 h. $n = 3$ or 6 independent experiments in **e–g** or **h**, respectively. Data were analyzed using one-way ANOVA followed by LSD post hoc test. Data are presented as mean values $+/-$ SEM. Source data are provided as a Source data file for **e–g**. **e** Glucose uptake assay using glucose analog 2-deoxyglucose (2-DG) to detect and quantify glucose uptake in chondrocytes. **f** Detection of total levels of cellular ATP based on the production of light caused by the reaction of ATP with added firefly luciferin. **g** Quantitative analysis of the ratio of pHrodo+ cells to total chondrocytes of pHrodo fluorescence staining in chondrocytes treated with low or high concentrations of recombinant TGFβ1. **h** Quantitative analysis of the ratio of DHE-positive cells to total chondrocytes treated with low or high concentration of recombinant TGFβ1.

conditional medium increased significantly in the mechanical stress+ group, and this phenomenon was not observed in the αV integrin knockdown group (Fig. 3h). Immunofluorescence staining of pSmad2 and Smad2 showed that the levels of pSmad2 or Smad2 nuclear translocation in chondrocytes were enhanced by subjecting the cells to shear stress, and knockdown of αV integrin abolished the effects of shear stress (Fig. 3i and Supplementary Fig. 5). The expression of αV integrin in chondrocytes was positively correlated with the magnitude of shear stress, with the peak occurring at a speed of 6.58 dynes/cm² (Fig. 3j). Similarly, the levels of pSmad2 continuously increased along with the increase in shear forces, even when the level of αV reached a plateau (Fig. 3j). Recombinant TGFβ1 treatment stimulated the expression of αV integrin and inhibition of TGFβ signaling. TGFβ type-I receptor inhibitor SB-505124 eliminated both TGFβ- and mechanical stress-stimulated αV integrin expression (Fig. 3k), indicating the existence of a positive feedback loop between αV integrin expression and TGFβ activation under mechanical stimulation.

Talin-centered cytoskeleton reorganization induces αV integrin-mediated TGFβ activation under mechanical stress. We further investigated the molecular mechanism of αV integrin-mediated TGFβ activation under mechanical stress. Chondrocytes isolated from mouse AC were cultured in the free well or under 6.58 dynes/cm² of shear stress. Most of the free well-cultured chondrocytes were in a spherical/polygonal shape, whereas most of the chondrocytes challenged by shear stress were in a dendritic/spindle-like shape (Supplementary Fig. 6). Immuno-fluorescence staining showed that the F-actin fibers were loosely piled up without dominant direction and relatively thin in the primary chondrocytes in the free well. Immunohistological staining of human AC showed that talin was expressed by the chondrocytes of both OA-M and OA-S specimens with slightly increased expression in the OA-S specimen (Supplementary Fig. 7). In the chondrocytes challenged by shear stress, the stress fiber formed with talin clustered at the tip of the bundles of actin filaments (Fig. 4a). Furthermore, immunoprecipitation (IP) demonstrated that the β6 subunit of the integrin complex binds to talin at a very low level in the chondrocytes cultured in free well, whereas mechanical stress enhanced the binding (Fig. 4b). The binding between αV integrin and LAP is the prerequisite for the αV integrin-mediated TGFβ activation. To further test whether talin binding to the β6 subunit regulates binding between αV integrin and LAP, we knocked down the expression of talin in chondrocytes with or without mechanical stress. The IP experiment showed that the binding of αV integrin to LAP was promoted by mechanical stress, and the effect was abolished by knocking down talin (Fig. 4c). Similarly, inhibition of polymer-ization of actin filament with cytochalasin D also abolished mechanical stress-induced binding of αV integrin with LAP

(Fig. 4d), suggesting that talin and actin polymerization are essential for binding of αV integrin with LAP. Indeed, western blot analysis showed that pSmad2/3 levels increased in the mechanical stress group and that knockdown of talin abolished the effect (Fig. 4e). A similar result was achieved by inhibition of polymerization of actin filaments using cytochalasin D (Fig. 4f). To validate that talin–integrin binding or actin polymerization under mechanical stress induces activation of TGFβ, we used enzyme-linked immunosorbent assay (ELISA) to measure the levels of active TGFβ in the conditional medium of chondrocyte culture with or without mechanical stress. Mechanical stress stimulated the production of active TGFβ, and this effect was abolished by knocking down talin or treating with cytochalasin D (Fig. 4g, h). We further tested the role of talin in mediating mechanical stress-induced TGFβ activation in the OA mouse model. We injected the siRNA of talin into the knee joint synovial capsule of mice received ACLT or sham surgery and harvested the knee joints of the mice 1 month post-surgery. Immunohis-tological staining of talin confirmed that the expression of talin in articular chondrocytes was successfully inhibited (Fig. 4i top, Fig. 4j). Importantly, the TGFβ signaling was significantly down-regulated by suppressing the expression of talin as evidence by a reduced number of pSmad2+ chondrocytes in the ACLT group (Fig. 4i bottom, Fig. 4k). This phenomenon was not observed in the sham-operated mice which indicates that talin particularly plays an essential role in the process of αV integrin-mediated TGFβ activation under mechanical stress. The safranin O fast green staining and OARSI grading showed that talin inhibition partially prevented cartilage degeneration that was induced by abnormal mechanical loading (Supplementary Fig. 8).

**Mechanical stress enhanced the traction force between αV integrin and its ligand**. To investigate the mechanism of talin–αV integrin in activation of TGFβ in response to mechanical stress, we used a double-stranded deoxyribonucleic acid (dsDNA) tether as a tension gauge to measure the tension tolerance of bonding of αV integrin with its RGD-containing ligands, such as latent TGFβ, in chondrocytes with or without mechanical stress. dsDNA has been used to measure the single molecular force because its rupture force can be precisely calculated according to the number/type of base pairs and force application geometry[53]. By conjugating biotin at different locations on the strand that was designed to bind the avidin on the polyethylene glycol (PEG) surface, dsDNA tethers with "tension tolerances" (rupture forces) of 43 and 54 pN were generated, respectively (Fig. 5a). The RGDfk was conjugated with the antisense strand of the dsDNA serving as a ligand for αV integrins on the chondrocytes that were seeded on top of the dsDNA tethers. When the contractile force generated by chondrocytes is higher than the tension tolerance of the dsDNA tether (tension gauge tether [TGT]), the dsDNA

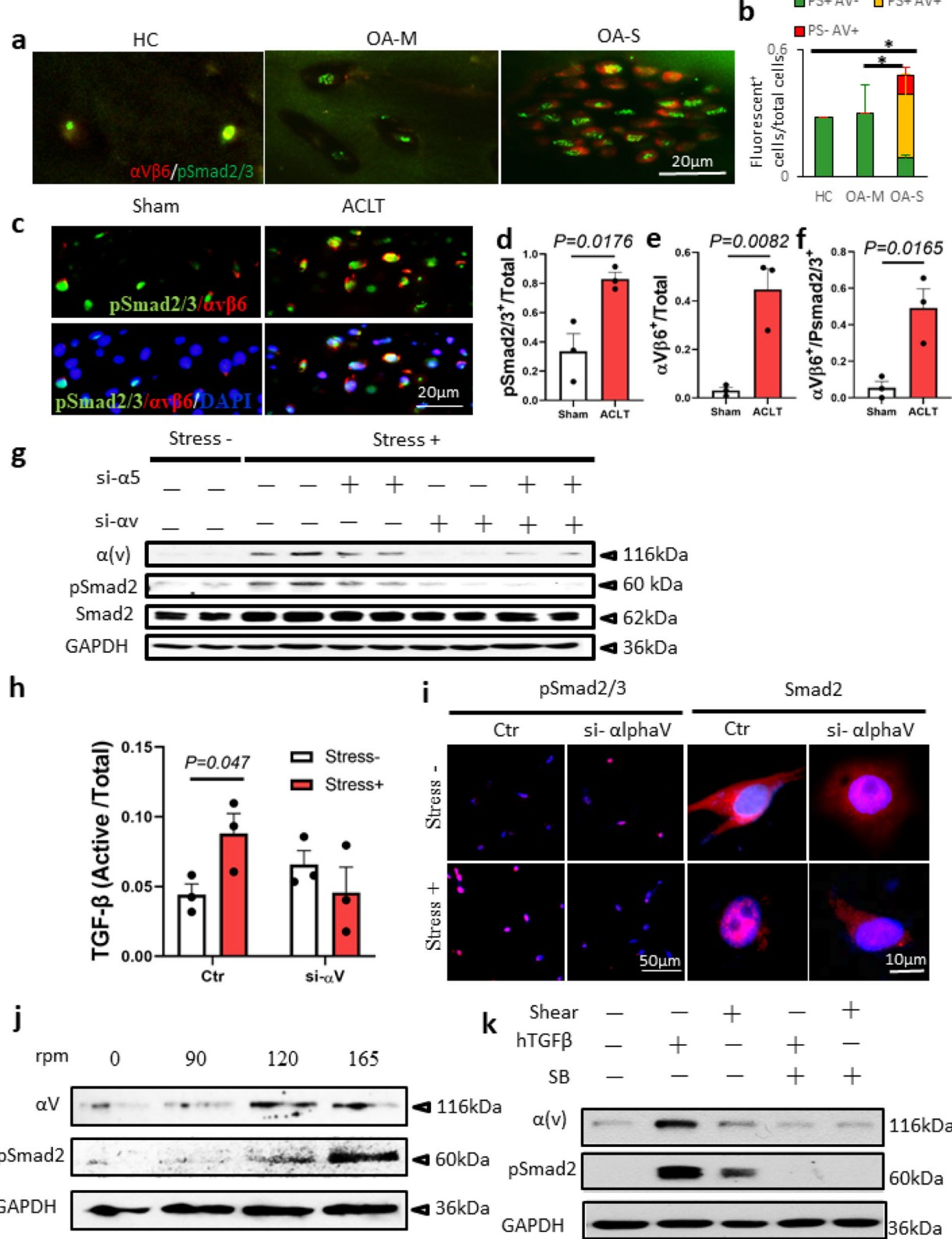

ruptures and a "black dot" is observed under the epi-fluorescence microscope because of the loss of Cy3 fluorescence at the rupture site. When most TGTs that link the chondrocytes with the PEG surface are ruptured, the attachment of the cell is lost (Fig. 5b, d). On both 43- and 54-pN TGT surfaces, chondrocytes in the shear stress group (shear+) induced TGT rupture more frequently

relative to chondrocytes in the shear− group. On the 43-pN TGT surface, the number of remaining chondrocytes in shear+ groups was significantly lower than that of the shear− groups, whereas the talin siRNA-treated shear+ group was comparable to the shear− groups. This result indicates that the contractile force that is exerted by most of the chondrocytes in the shear+ group

**Fig. 3 αV integrins mediate mechanical stress-induced TGFβ activation in chondrocytes. a** Double immunofluorescence staining of αVβ6 (red) and pSmad2/3 (green) in AC of human knee joints. HC healthy control, OA-M OA specimen with minimal cartilage degeneration, OA-S OA specimen with severe cartilage degeneration. **b** Quantitative analysis of αVβ6 and pSmad2/3 double staining in **a**. $n = 10$ biologically independent specimens, *$P < 0.05$ in comparison to αVβ6 and pSmad2/3 double-positive cells. Data are presented as mean values $+/−$ SEM. Data were analyzed using one-way ANOVA LSD post hoc test. PS+: pSmad2/3+, AV+: αVβ6+. **c** Immunofluorescence staining of αVβ6 (red) and pSmad2/3 (green) in mouse AC harvested 1 month after ACL-T or sham surgery. Nuclei were labeled with DAPI (blue). **d–f** Quantitative analysis of the percentage of pSmad2/3+ cells (**d**) and αVβ6+ cells (**e**) in total chondrocytes and the ratio of αVβ6+ cells in pSmad2/3+ cells (**f**). $n = 3$ biologically independent animals, data are presented as mean values $+/−$ SEM. Data were analyzed by two-tailed unpaired $t$ test. **g** Western blot of αV integrin and pSmad2 protein in SV40 chondrocyte cell line. The cells were cultured in the free well or subjected to shear stress (6.58 dynes/cm$^2$) for 48 h. The gene expression of αV integrin or α5 was knocked down using siRNA. Source data are provided as a Source data file. **h** ELISA of TGFβ in the conditional medium of the SV40 chondrocyte culture. The cells were cultured in the free well or subjected to shear stress (6.58 dynes/cm$^2$) for 48 h. $n = 3$ independent experiments, data are presented as mean values $+/−$ SEM. Data were analyzed using one-way ANOVA LSD post hoc test. Ctr or si-alphaV: cells treated with siRNA of control or αV integrin, respectively. **i** Immunofluorescence staining of pSmad2/3 and Smad2 in the SV40 cells with or without shear stress. Ctr or si-alphaV cells treated with siRNA of control or αV integrin, respectively. **j** Western blot of αV integrin and pSmad2 protein in SV40 chondrocyte cell line subjected to shear stress at different speeds. **k** Western blot of αV integrin and pSmad2 protein in SV40 chondrocyte cell line. Shear shear stress, hTGFβ recombinant human TGFβ1 (2 ng/ml), SB SB505124, TβRI selective inhibitor (1 μM). All the in vitro experiments were repeated three times independently. Source data are provided as a Source data file.

was ≥43 pN, the minimal required force required to induce conformational change of LAP in activation of TGFβ, and that knocking down the expression of talin abolished the effect of mechanical stress (Fig. 5c). On the 54-pN TGT surface, although the remaining chondrocyte numbers increased in all groups compared with that on 43-pN TGT surface, the cell numbers in shear+ groups were still significantly lower than those of the shear− groups, suggesting that a substantial number of chondrocytes in the shear+ group can exert a contractile force ≥54 pN (Fig. 5c). Similarly, talin knockdown significantly reduced the rupture of 54-pN TGT (Fig. 5d), although it did not reach the threshold for disturbing the cell attachment (Fig. 5c). Because the contractile force exerted by the chondrocytes was relayed by the αV integrin–RGD bond, our result suggests that the αV integrin–RGD bond could bear a strong molecular tension >43 pN and even 54 pN in chondrocytes that are challenged by mechanical stress.

To further investigate the mechanism by which mechanical stress enhances the tension tolerance of the integrin–RGD bond, we examined chondrocyte mechanics probed with live cell micromechanical methods and traction microscopy (Supplementary Fig. 9). Consistent with the findings in the TGT experiment, chondrocytes subjected to shear stress showed greater root mean square (RMS) traction forces compared with that of the no-stress group. Silencing talin expression with siRNA abolished the effect of shear stress in increasing the traction forces (Fig. 5e–g). Moreover, the chondrocytes subjected to shear stress had significantly greater stiffness, and knocking down talin attenuated the increase of stiffness induced by shear stress (Fig. 5h). Shear stress also reduced the rate of cytoskeletal remodeling, and this reduction was not significantly altered when talin was silenced (Fig. 5i and Supplementary Fig. 10). To validate the mechanism in an OA animal model, we performed live cell biomechanical testing in chondrocytes isolated from the AC of OA mice that underwent ACL-T or sham surgery. As indicated by FEA, the anterior region of the tibial plateau experienced greater mechanical stress relative to that of the posterior region in ACL-T mice. We found that the cell stiffness of chondrocytes from the anterior region was significantly greater than that of the posterior region in OA mice. In contrast, the cell stiffness did not differ between chondrocytes collected from different regions of sham-operated mice (Fig. 5j). Taken together, these data demonstrate that shear stress increases cell stiffness, enhances the contractile force of chondrocytes, and increases the tension tolerance of the integrin–RGD bond in a talin-dependent manner. It is conceivable that TGFβ activity will increase in the AC region when a significant number of chondrocytes in that region exert a

contractile force above the minimum contractile force requirement for TGFβ activation. Our finding at least partially explains the uneven TGFβ activity distribution in osteoarthritic AC under aberrant mechanical loading.

**αV integrin-mediated excessive TGFβ activation for AC degeneration in OA mice.** To validate the potential role of αV integrin in excessive TGFβ activation in an OA mouse model, we crossed col2aCre$^{ERT}$ mice with αV$^{fl/fl}$ mice to generate col2a-Cre$^{ERT}$:αV$^{fl/fl}$ (αV$^{−/−}$) mice, in which *ITGAV* is specifically deleted in type-II collagen (Col2a)-expressing cells (chondrocytes) upon tamoxifen injection. The structural difference in SB between αV$^{−/−}$ ACLT OA mice and sham-operated mice was similar to that between wild-type ACLT OA mice and sham-operated mice, indicating that deletion of the *ITGAV* in chondrocytes did not affect the structure of SB (Fig. 6a). However, deletion of *ITGAV* counteracted the effect of abnormal mechanical stress in regionally elevating TGFβ activity in AC. The distribution and number of pSmad2/3+ cells in AC of αV$^{−/−}$ OA mice was similar to that of both sham-operated αV$^{−/−}$ and wild-type mice (Fig. 6b). Moreover, the proteoglycan loss at the anterior region was attenuated in αV$^{−/−}$ mice after ACL-T surgery, and the proteoglycan levels in the αV$^{−/−}$ ACL-T group were comparable to those of the sham-operated αV$^{−/−}$ or wild-type groups (Fig. 6c). Consistently, immunofluorescence staining of pSmad2/3 showed that conditional deletion of *IGTAV* in chondrocytes abolished the aberrant activation of TGFβ in anterior AC, the region that has been found to have the higher mechanical stress in FEA (Fig. 6c). Double immunostaining of pSmad2/3 with aggrecan showed that the number pSmad2/3+ chondrocytes in the anterior region significantly increased in OA mice relative to sham-operated mice. Notably, most of the pSmad2/3+ cells were negative for aggrecan staining in wild-type ACL-T mice (Fig. 6d, f). The number of pSmad2/3+/aggrecan− cells were significantly decreased and the pSmad2/3−/aggrecan+ cells were increased in the cartilage, particularly in the anterior region, when the *αV* gene was conditionally deleted in chondrocytes (Fig. 6d, g). Co-immunofluorescence staining of αVβ6 and vimentin, a marker for fibrochondrocytes, showed that the number of αV/vimentin double-positive chondrocytes was significantly increased in the AC of ACL-T mice relative to that of sham-operated mice (Supplementary Fig. 11a). The ratio of 8-OHdg+ chondrocytes was increased in the AC of wild-type mice after ACL-T. Importantly, most 8-OHdg+ chondrocytes were also positive for pSmad2/3. The increase of 8-OHdg+/pSmad2/3+ chondrocytes in AC after ACL-T was prevented by conditionally deleting the *αV* gene (Fig. 6e, h-i). Considering that 8-OHdg

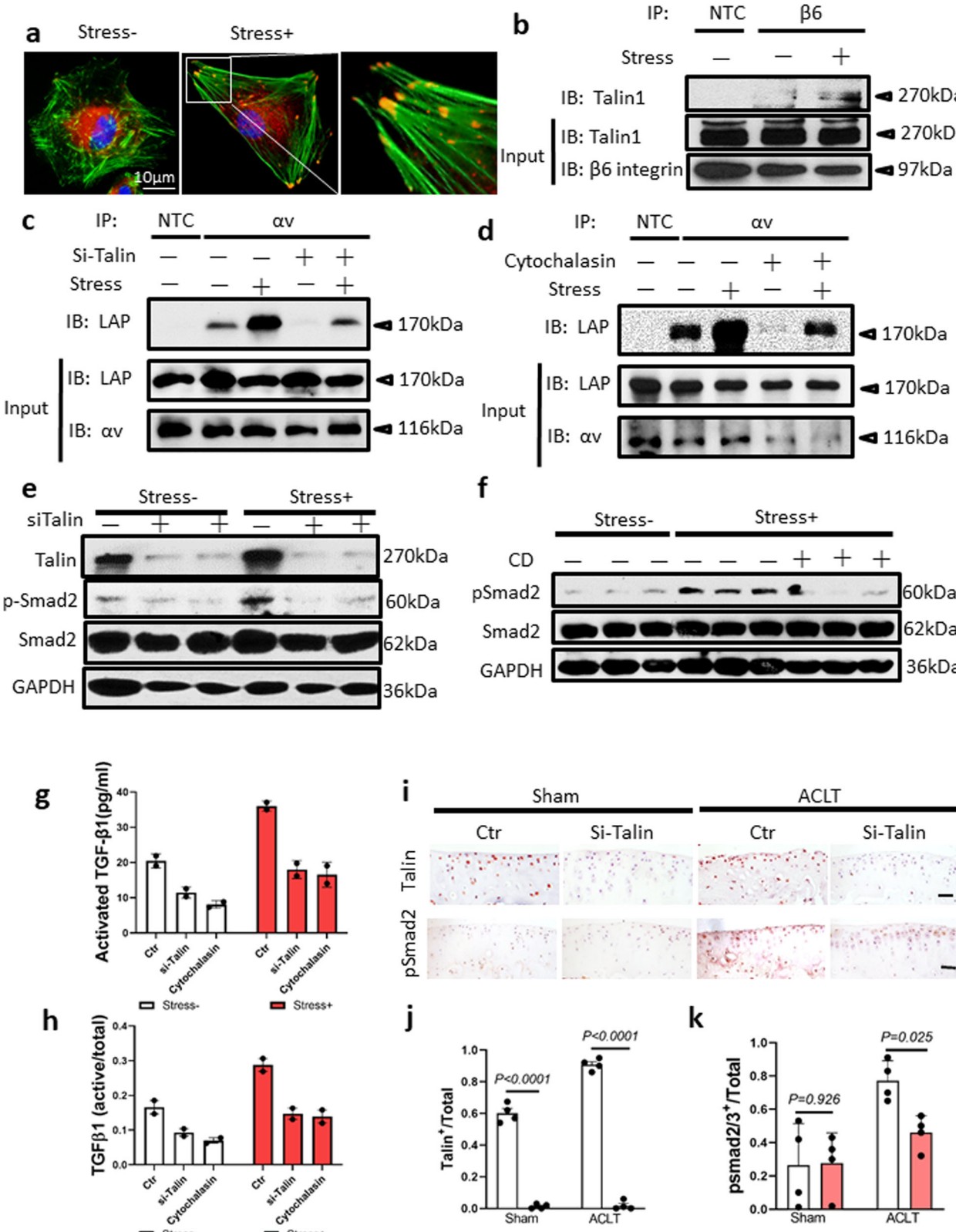

labels the oxidative modifications of DNA, our findings indicate that αV⁺-induced excessive TGFβ activation is associated with free radical-induced DNA damage in chondrocytes, which may further contribute to decreased aggrecan synthesis and fibro-cartilage formation. We further validated the findings by incubating primary chondrocytes in different concentrations of recombinant TGFβ1. Aggrecan production indeed decreased in the chondrocytes that were cultured in a high concentration of TGFβ1 (Supplementary Fig. 11b). Moreover, a high level of active TGFβ increases chondrocyte stiffness in chondrocytes with or without shear stress measured by magnetic twisting cytometry (Supplementary Fig. 11c), suggesting that excessive TGFβ impairs

**Fig. 4 Talin-centered cytoskeleton reorganization transmitted mechanical stress to αV integrins in activation of TGFβ. a** Immunofluorescence staining of talin (red) in primary chondrocytes isolated from mouse AC. F-actin was labeled by phalloidin (green). Primary chondrocytes were subjected to shear stress (6.58 dynes/cm²) for 48 h. **b–d** IP experiments. IP was carried out using the lysates of the SV40 cells subjected to 48-h shear stress or cultured in the free well. IP immunoprecipitation, IB immunoblotting, NTC negative control, β6 β6 integrin, αV αV integrin, si-Talin talin siRNA. Source data are provided as a Source data file. **e, f** Western blot of the SV40 cell lysates. The cells were subjected to shear stress or cultured in the free well for 48 h. CD cytochalasin D. Source data are provided as a Source data file. **g, h** ELISA of the concentration of active form TGFβ (**g**) or ratio to total TGFβ (**h**) in the conditional medium of the SV40 chondrocyte culture. The cells were cultured in the free well or subjected to shear stress for 48 h. n = 2 independent experiments. Data are presented as mean values +/− SEM. Source data are provided as a Source data file. **i** Representative image of immunohistochemical staining of talin (top, brown) and pSmad2 (bottom, brown) of AC of knee joints harvested from C57BL/6 mice subjected to and intra-articular injection of si-Talin or control siRNA for 1 month post ACLT or sham surgery. Scale bar: 50 µm. **j, k** Quantitative analysis of talin staining (**j**) and pSmad2 staining (**k**). n = 4 biologically independent animals. Data are presented as mean values +/− SEM. Data were analyzed using one-way ANOVA LSD post hoc test.

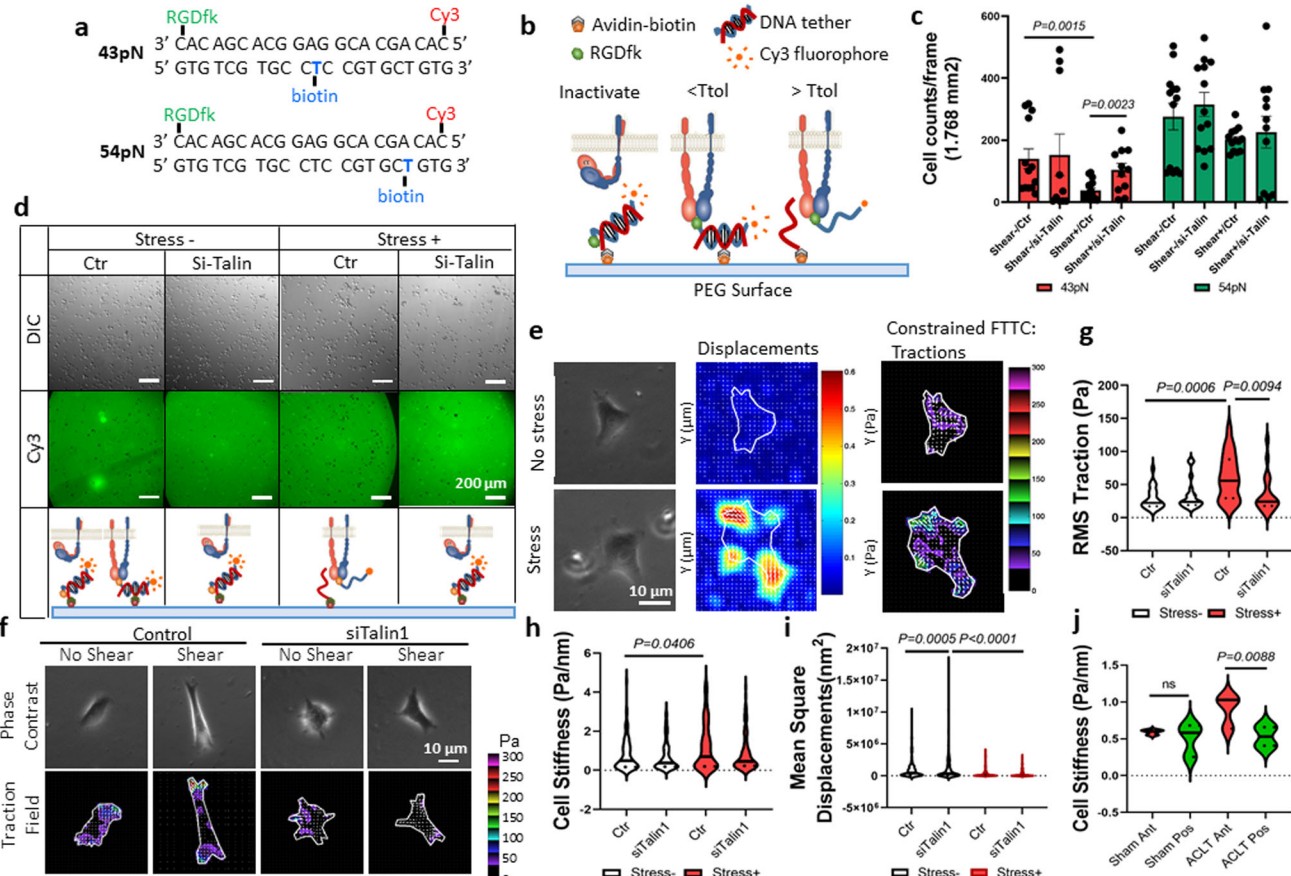

**Fig. 5 Talin mediates the strengthening of αV integrin–RGD bond under mechanical stress. a** The sequence of the dsDNA and the conjugation sites of specific molecules. **b** Schematic illustration of the TGT principle. A dsDNA tether is immobilized on the PEG surface through an avidin–biotin linker. The location of the biotin determines the tension tolerance (Ttol) of the dsDNA. The integrins expressed by the cell bind to the RGDfk conjugated with the dsDNA at one end. The dsDNA ruptures if the tension applied by the cell through the integrin–RGD bond is greater than its Ttol. The Cy3 fluorescence signals and the adhesion of the cells on the PEG surface are maintained if the tension applied by the cell through the integrin–RGD bond is lower than its Ttol. **c** Cell density represents the rupture of the DNA tether. n = 13-17 independent cells. Data are presented as mean values +/− SEM. Data were analyzed using two-tailed t test. Source data are provided as a Source data file. **d** Representative differential interference contrast (DIC) images of the SV40 cells on the PEG surface (upper row) and the direct imaging of RGDfK-dsDNA-Cy3 removal on the PEG surface (bottom row). The cells were prechallenged with or without shear stress. The quantitative analysis is shown in **c**. Ctr cells were treated with control siRNA, Si-Talin cells were treated with talin siRNA. **e** Representative phase contrast (left column), the displacement of magnetic beads (middle column), and traction map images (right column) of SV40 cells that were preconditioned with or without shear stress. White lines show the cell boundary; colors show the magnitude of the tractions in Pa (see color scale). **f** Representative phase contrast (upper row) and traction map images (bottom row) of SV40 cells preconditioned with or without shear stress. Ctr cells were treated with control siRNA, Si-Talin cells were treated with talin siRNA. The quantitative analysis results are shown in **g–i**. **g–i** Quantitative analysis of RMS traction force (**g**), cell stiffness (**h**), and cytoskeleton remodeling (**i**) based on the displacement of magnetic beads. n = 10 independent experiments. Data were analyzed using two-way ANOVA Turkey's post hoc tests. Source data are provided as a Source data file. **j** Quantitative analysis of cell stiffness of primary chondrocytes isolated from AC of mice tibia medial compartment. The mice were sacrificed 1 month after ACL-T or sham surgery. Ant anterior region, Pos posterior region. n = 3 independent animals. Data were analyzed using two-way ANOVA Turkey's post hoc test. Source data are provided as a Source data file.

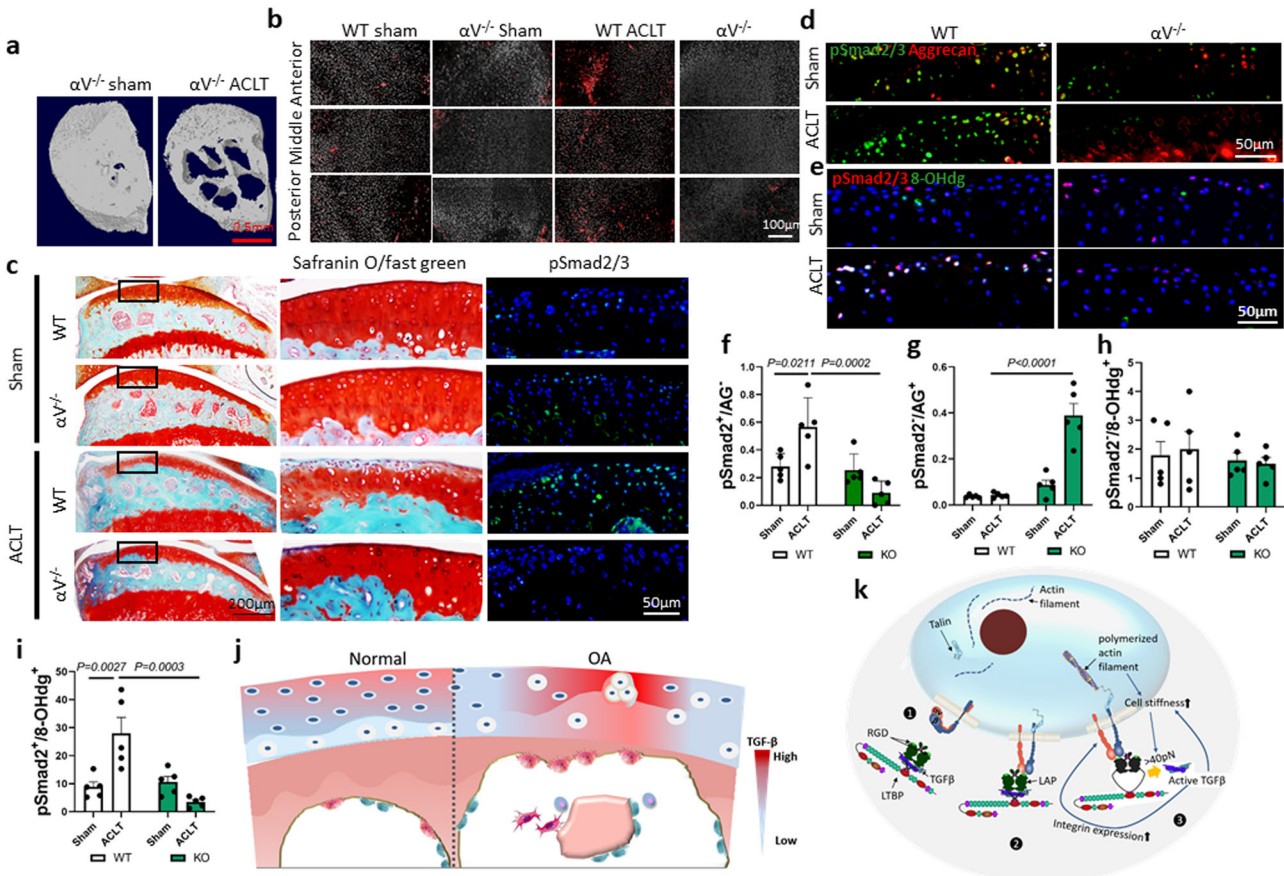

**Fig. 6 αV integrin-mediated excessive TGFβ activation in AC degeneration of OA mice. a** Transaxial view of 3D reconstructed images of tibia medial compartment from Col2a-Cre^ERT: αV^fl/fl (αV^−/−) mice at 1 month after ACL-T or sham surgery. **b** Representative immunofluorescence staining images of pSmad2/3 in AC isolated from mice tibia medial compartment at 1 month after ACL-T or sham surgery (red: pSmad2/3, gray: DAPI). n = 3 biological independent animals in **a**, **b**. **c** Safranin O/fast green staining (left and middle columns) and fluorescence staining of pSmad2/3 (right column) of the sagittal sections of knee joint medial compartment of mice sacrificed at 1 month post sham or ACLT surgery. pSmad2/3: green, DAPI: blue. n = 3 biological independent animals. **d–i** Double immunofluorescence staining of pSmad2/3 (green) and aggrecan (red) in **d** and pSmad2/3 (red) and 8-OHdg (green) in **e**, and the AC were collected from mice at 1 month after ACL-T or sham surgery. **f** The ratio of pSmad2/3-positive and aggrecan-negative chondrocytes in total chondrocytes. **g** The ratio of pSmad2/3-negative and aggrecan-positive chondrocytes in total chondrocytes. **h** The ratio of pSmad2/3-negative and 8-OHdg-positive chondrocytes in total chondrocytes. **i** The ratio of pSmad2/3 and aggrecan double-positive chondrocytes in total chondrocytes. **f–i** n = 5 biologically independent animals. Data were analyzed using two-way ANOVA followed by Turkey's post hoc analysis. Data are presented as mean values +/− SEM. **j** Schematic illustration of aberrant configuration of the TGFβ activity in OA cartilage due to the redistribution of mechanical stress in AC. **k** Schematic illustration of abnormal mechanical stress induced aberrant TGFβ activation and subsequent influence on chondrocyte metabolism in OA articular cartilage. Structural alteration of subchondral bone disrupts the mechanical environment in above cartilage. Under low mechanical stress, the αV integrin is in an inactive status and the latent TGFβs are deposited in the extracellular matrix. High mechanical stress induces the activation of αV integrins and the reorganization of cytoskeleton (stress fiber formation). Stress fiber formation increases cell stiffness and ensures chondrocytes exert contractile force higher than the minimal forces required for a conformational change of LAP and release of active TGFβ. Upregulated TGFβ signaling in turn increases cell stiffness and αV integrin expression in chondrocytes.

the synthesis of matrix proteins in chondrocytes. In the meantime, it also increases cell stiffness, which further facilitates cell contractile force-induced TGFβ activation.

Because deletion of αV prevented excessive TGFβ activation and consequent cartilage degeneration in OA mice, we tested whether recombinant RGD peptide could act as a neutralizing ligand for αV integrin to prevent AC degeneration in OA mice. Indeed, degeneration of AC has been partially improved by RGD intra-articular injection in OA mice relative to vehicle-treated mice. Unfortunately, mild proteoglycan loss was also observed in RGD-treated sham-operated mice, possibly because of the nonspecific binding of RGD peptide to other domains in chondrocytes or suppressing the normal TGFβ activation (Supplementary Fig. 12). Thus, intra-articular injection of RGD

can partially mimic the beneficial effect of the conditional deletion of the αV gene in osteoarthritic cartilage.

## Discussion

TGFβ has been considered an anabolic factor essential for AC homeostasis. However, both deleterious and protective effects of TGFβ have been observed on AC in human OA and animal models. Both increased and decreased levels of TGFβ activity have been associated with osteoarthritic cartilage[18–21,27,54,55]. Our findings support the concept that temporal–spatial activation of TGFβ to maintain TGFβ activity within an appropriate range is essential for the maintenance of AC metabolic homeostasis and structural integrity; activity above or below the range leads to AC degeneration. We found that TGFβ signaling is not universally

upregulated or downregulated in osteoarthritic cartilage. Because of aberrant mechanical loading, the distribution of TGFβ activity in osteoarthritic AC is altered—increased in high-stress regions and decreased in low-stress regions. In either situation, TGFβ activity outside the normal physiological range is detrimental to chondrocyte metabolism and cartilage structural integrity. The uneven distribution of TGFβ induced by abnormal mechanical stress may explain the multiple phases of cartilage degeneration observed in the osteoarthritic joint. The alteration of the expression of TGFβ receptors in chondrocytes has been observed in various OA animal models[56,57]. It is conceivable that the altered expression of TGFβ receptor amplifies the detrimental effect of regional excessive TGFβ activation in aged or osteoarthritic AC.

AC degeneration is the primary concern in OA, yet its homeostasis and integrity rely on the biochemical and biomechanical interplay with SB and the other joint tissues[12]. Our study demonstrated that the distribution of TGFβ activity in AC is well aligned with the structural configuration of SB. We previously showed that aberrant activation of TGFβ during osteoclastic bone resorption at the onset of OA disrupts the migration of mesenchymal stem cells and leads to pathological changes in SB[58,11]. In this study, we demonstrated that SB structural alterations lead to redistribution of mechanical stress in AC, which in return determines the pattern of TGFβ activation in AC. Indeed, AC and SB are integrated through the osteochondral junction and act as a functional unit[59,60]. It has been reported that small molecules such as sodium fluorescein could diffuse readily from SB to the articular space[61,62]. Moreover, neovasculature that penetrates from SB to AC also facilitates the biochemical cross-talk between these two tissues[63]. However, the tidemark and mineralized cartilage may not be permeable to larger molecules, such as TGFβ. The dense, nonspecific TGFβ-binding sites also restrict the free diffusion of TGFβ within AC[64]. The abundant storage of latent TGFβ provides the material base for local activation of TGFβ in AC. Because of its greater stiffness and strength compared with the overlying AC[65–67], SB act as a structural girder and shock absorber during AC–SB load transmission. The effect of structural changes in the SB on stress/strain distributions in the AC has been demonstrated and explored in detail in previous studies[68]. It was shown that, by creating local stiffening (densification) in the SB that mimics an osteoarthritic bone phenotype, stresses in the overlaying cartilage could be elevated by up to 50%. The explanation for this observation was that sclerotic SB has reduced ability to attenuate and distribute load evenly throughout the joint, which subsequently increases stress and creates uneven stress distributions in the overlaying AC, as the cartilage and bone work together in load-bearing in the joint. In the present study, local increased BV/TV and reduced structure model index (SMI; indicating a more plate-like structure) are signs that the SB has become sclerotic, which is stiffer and more unevenly distributed. Our finding of TGFβ activation by mechanical stress in AC uncovers the mechanism of how SB controls the configuration of TGFβ activity in AC.

The mechanism of chondrocyte mechanotransduction (i.e., how mechanical forces are transformed into biochemical signals in chondrocytes) is under active investigation. We show that chondrocytes sense and respond to abnormal mechanical signals through the αV integrin and talin-centered cytoskeletal system. The unique feature of αV-containing integrins to bind to the RGD sequence in the LAP allows αV integrin to mediate TGFβ activation through cell contractile forces. Mechanical stress strengthens the bond between αV integrin and RGD and increases its tension tolerance. The increased cell stiffness and elevated tension tolerance of the αV integrin–RGD bond prepared chondrocytes to withstand the contractile forces required

by TGFβ activation. Furthermore, as one of the key adapter proteins in the system of cellular mechanotransduction, talin establishes mechanical linkages between the cytoskeleton and the integrin heterodimers[69–71]. We found that mechanical stress promotes the binding of talin to the cytoplasmic domain of the integrin, as well as to the F-actin bundles. When talin expression was inhibited, the increase of cell stiffness and elevated contractile forces between αV integrin and the RGD sequence in response to mechanical stress was diminished. Talin is an important cytoskeletal protein and global knockout of talin is embryonic lethal due to early developmental abnormalities[72]. Studies employing the conditional knockout of talin 1 in different tissues showed variety of pathological alterations, such as progressive myopathy[73] and severe hemostatic defects[74]. Although conditional deletion of talin gene (TLN1) in chondrocytes may blunt the excessive TGFβ activation under mechanical stress, the overall in vivo outcome is hard to predict as the absence of talin in chondrocytes itself may lead to functional or structural impairment of AC. Consequently, the mechanical loading-induced TGFβ activation that was mediated by αV integrin was abolished. This finding indicates that talin has a key role in the mechanosensory system of chondrocytes, which coordinates cytoskeleton reorganization while reinforcing integrin–RGD interactions. Our study provides a possible explanation for how extracellular mechanical signals are translated into intracellular mechanical signals and thereby further modify the cell fate and behaviors.

The integrin-mediated contractile force activates latent TGFβ in a nonlinear mechanism, which occurs only when contractile forces reach a certain level (approximately 40 pN). This mechanism also explains why TGFβ activity significantly decreased in AC areas with less mechanical stress in human OA specimens and animal models. Moreover, TGFβ promotes the expression of αV integrins and increases chondrocyte stiffness through a positive feedback mechanotransduction cascade. Importantly, disruption of the "on/off" switch for TGFβ activation in a high or low mechanical stress environment amplifies the cascade of detrimental effects and results in cartilage fibrillation during the early stage of OA. Therefore, modulation of αV integrin expression in AC could be a potential therapeutic target for OA. Targeting αV integrins may have the advantage of locally eliminating the detrimental effects of excessive TGFβ in AC without disturbing cartilage homeostasis by further reducing TGFβ activity in areas of low mechanical stress. Normalization of mechanical loads on joints through rescuing SB bone remodeling may prevent cartilage degeneration during the early stage of OA.

## Methods

**Human subjects**. After institutional review board (IRB; Johns Hopkins Institutional Review Boards) approval, we collected tibial plateau specimens from 78 patients with OA who were undergoing total knee replacement. The waiver of the consent from the participants was approved by our institutional IRB because the specimens are de-identified tissue archived by the pathological department, which meet the US Food and Drug Administration regulation of consent waiver (Organization Policy FDA 50.1). The specimens were processed for μCT and histological examination. The OA specimens were collected from the tibial plateau specimens of OA patients who received total knee replacement. The specimens were examined by raw eyes and one OA-M and one OA-S sample at $1 \times 1 \times 1$ cm$^3$ were collected from each OA specimen. The samples with intact cartilage coverage under the raw eye were categorized into the OA-M group while samples with severe cartilage damage were categorized into the OA-S group. The healthy knee specimens were purchased from the National Disease Research Interchange (Philadelphia, PA) to serve as controls. These knee joint specimens are collected from human donors without a history of OA.

## Animals

*Mice*. We purchased C57BL/6J (wild-type) mice from Charles River Laboratories (Wilmington, MA). We anesthetized 3-month-old male mice with ketamine and xylazine and then transected the anterior cruciate ligament surgically to induce mechanical instability-associated OA on the right knee. Sham operations were

performed on independent mice. Operated mice were euthanized at 30 days postoperatively, $n = 8–12$.

We purchased *Col2a1–CreERT* mice (strain name: FVB-Tg(Col2a1-cre/ERT) KA3Smac/J) from the Jackson Laboratory (Bar Harbor, ME). Floxed αV integrin receptor (αV$^{fl/fl}$) mice were obtained from Dr. Lacy-Hulbert's laboratory (Benaroya Research Institute, Seattle, WA). *Col2a1–CreERT* mice were crossed with αV$^{fl/fl}$ mice. The offspring were intercrossed to generate the following offspring: *Col2a1–CreERT::* αV$^{fl/fl}$ mice, in which *Cre* was fused with a mutated estrogen receptor that could be activated by tamoxifen. We determined the genotype of transgenic mice by PCR analysis of genomic DNA isolated from mouse tails. Genotyping for the Cre transgene was performed by PCR with the primers Cre 5′ (5′-CACTGCGGGCTCTACTTCAT-3′) and Cre 3′ (5′-ACCAGCAGCACTTTTGGAAG-3′). The *loxP* αV allele was identified with the primers lox1F (5′-GGTGACTCAATGTGTGACCTTCAGC-3′) and lox1R (5′-CACAAATCAAGGATGACCAAACTGAG-3′) (Supplementary Table). We performed ACL-T or sham surgery on 3-month-old, male *Col2a1– CreERT:*αV$^{fl/fl}$ mice; αV$^{fl/fl}$ littermates were used as wild-type controls. One week before surgery, we treated each group with 80 mg/kg bodyweight of tamoxifen twice a week for 37 days and euthanized the mice at 30 days postoperatively ($n = 8$ per treatment group). The dosage and treatment frequency were determined based on our preliminary experiment.

*Intra-articular injection of siRNA solution.* Three-month-old C57BL/6 mice were randomly distributed into four groups: sham group treated with control siRNA, sham group treated with talin siRNA, ACLT group treated with control siRNA, and ACLT group treated with talin siRNA. The talin siRNA or control RNA were prepared by dissolving 5 nM siRNA (Ambion, Austin, TX) in 1 ml TransIT®-QR Delivery Solution (Mirus, Madison, WI). Starting from 1 day post surgery, 30 μl ($10^{-2}$ nM) of talin siRNA or control siRNA were injected into the synovial capsule of the mice knee joints once a week under sterile condition using a microliter syringe. The mice knee joints were harvested at 28 days after injection.

All animals were maintained in the animal facility of The Johns Hopkins University School of Medicine (Baltimore, MD). The experimental protocol was reviewed and approved by the Institutional Animal Care and Use Committee of The Johns Hopkins University.

**Cell culture**. We purchased immortalized human chondrocytes SV40 from Applied Biological Materials (Richmond, BC, Canada). We maintained cells (passage 3–10) in Iscove's modified Dulbecco's medium (Invitrogen, Carlsbad, CA) supplemented with 10% fetal calf serum (Atlanta Biologicals, Flowery Branch, GA) and 1% penicillin–streptomycin (Mediatech, Manassas, VA). We cultured SV40 cells in 6-well plates at a density of $1 \times 10^5$ cells per well for 7 days. The pure ECM secreted by SV40 chondrocytes was obtained by removing cells with ice-cold desoxycholate buffer (0.5% desoxycholate, 1% NP-40, and 150 mM NaCl in 10 mM Tris-HCl [pH 8.0]) for 10 min, followed by washing with phosphate-buffered saline. Then the second batch of SV40 cells was seeded on the ECM remaining plates at a density of $5 \times 10^4$/well. The cells were cultured in free wells in Dulbecco's Modified Eagle Medium (DMEM) with serum-free media supplement (Nutridoma-SP, Roche, Basel, Switzerland) for 48 h. The orbital shear stress was applied at 37 °C in the presence of $CO_2$ by placing the 6-well plates on an orbital shaker (Thermo multipurpose rotator, ThermoFisher Scientific, Waltham, MA). The shear stress across the cells on the periphery of the dishes was calculated as $\tau_{max} = a\sqrt{\rho\eta(2\pi f)^3}$, where $a$ is the radius of orbital rotation (1.75 cm), $\rho$ is the density of the medium (1.0 g/ml), $\eta$ is the viscosity of the medium ($7.5 \times 10^{-3}$ dynes·s/cm$^2$), and $f$ is the frequency of rotation (rotations/s)[75–78]. Using this equation, shear stress of 6.58 dynes/cm$^2$ is achieved at a rotating frequency of 118 rpm. After shear stress, only the cells on the periphery of the dishes were collected for further analysis. The cells that underwent siRNA treatment (Santa Cruz Biotechnology, Dallas, TX) were cultured in a 6-well plate at a density of $5 \times 10^4$. One day before transfection, culture medium was changed to DMEM without the supplement of antibiotics. Twenty minutes before transfection, siRNA was mixed with lipofectamine 2000 and held at room temperature for 20 min. The siRNA–lipofectamine 2000 mixture was then added to cultured cells at a final concentration of 50 nM. After 6 h of incubation, the transfection was terminated by replacing the medium that contained siRNA–lipofectamine 2000 with fresh culture medium, and the cells were collected for further analysis.

**Cell metabolism assays**. SV40 cells were seeded in a 6-well plate at a density of $5 \times 10^4$/well. The cells were cultured in DMEM supplemented with 10% fetal calf serum (Atlanta Biologicals), and 1% penicillin–streptomycin (Mediatech). Recombinant TGFβ1 (R&D Systems, Minneapolis, MN) was added to the culture medium at final concentrations of 2 and 10 ng/ml, respectively. Glucose uptake assay was performed using the Glucose Uptake Assay Kit (Abcam, Cambridge, UK) according to the manufacturer's instructions. ATP production was measured using the Luminescent ATP Detection Assay Kit (Abcam) following the protocol provided by the manufacturer. Intracellular pH detection was performed using the pHrodo Red AM intracellular pH indicator (Life Technologies, Carlsbad, CA). Ten-microliter pHrodo Red AM was mixed with 100 μl of PowerLoad concentrate and added to 10 ml of HEPES buffer. After washing the cells with HEPES buffer, the pHrodo Red AM/PowerLoad/HEPES mix was added to cells and incubated at

37 °C for 30 min. Then the cells were stained with 4,6-diamidino-2-phenylindole (DAPI; 1:1000, 15 min at room temperature) to label the nucleus. The pHrodo Red AM fluorescence was detected under a fluorescence microscope. The oxidative fluorescent DHE was used to evaluate the intracellular production of ROS. The cells were incubated in phenol red-free Hanks containing DHE (10 μmol/l) at 37 °C for 30 min. Then the cells were stained with DAPI (1:1000, 15 min at room temperature) to label the nucleus. DHE fluorescence was detected under the microscope.

**ELISA, western blot, and co-IP**. We determined the concentration of active TGF-β1 in the conditioned media by the ELISA Development Kit (R&D Systems) according to the manufacturer's instructions. We measured the levels of both active and total TGFβ using ELISA assay. As instructed by the vendor-provided manual, the active TGFβ was measured directly from the conditioned media while the total TGFβ was measured after treating with acid. IP was performed on the protein of lysates from in vitro cultured SV40 cells and their secreted ECM. The lysates were incubated with primary antibody (anti-ITGB6, Abcam, 1:50 or anti-ITGAV, Cell Signaling Technology, Danvers, MA, 1:50) at 4 °C under rotary agitation overnight. The antigen sample/antibody mixture was then incubated with pre-washed Protein A coupled Magnetic beads (ThermoFisher Scientific) under rotary agitation for 1 h at room temperature. After carefully washing and removing the nonspecific binding, the sample/antibody/beads conjugates were collected with a magnetic stand. The beads were eluted by adding 60 μl of sodium dodecyl sulfate–polyacrylamide gel electrophoresis (SDS-PAGE) reducing sample buffer and incubating the sample for 5 min at 100 °C. The supernatant containing the target antigen was saved for further western blot. Western blot analyses were conducted on the protein of lysates from in vitro cultured SV40 cells or the protein complex pulled down by co-IP. The cell lysates were centrifuged at a rcf of $180 \times g$, and the supernatants were separated by SDS-PAGE and blotted on polyvinylidene fluoride membrane (Bio-Rad Laboratories, Hercules, CA). After incubation in specific antibodies, we detected proteins using an Enhanced Chemiluminescence Kit (Amersham Biosciences, Little Chalfont, UK). We used antibodies recognizing human pSmad2 (Cell Signaling Technology, #3108, 1:1000), Smad2 (Cell Signaling Technology, #5339 S 1:1000), integrin αV (Cell Signaling Technology, #4711, 1:1000), LAP antibody (R&D Systems, MAB246-SP, 2 μg/ml), and anti-talin1 (Abcam, ab71333, 1 μg/ml) to examine the protein concentrations.

**Histochemistry and immunohistochemistry**. At the time of mouse sacrifice, we resected and fixed the knee joints in 10% buffered formalin for 48 h, decalcified in 10% ethylenediamine tetraacetic acid (pH 7.4) for 21 days, and embedded in paraffin or optimal cutting temperature compound (Sakura Finetek USA, Torrance, CA). Four-μm-thick sagittal oriented sections of the knee joint medial compartment were processed for hematoxylin and eosin and safranin O/Fast green staining. Tartrate-resistant acid phosphatase staining was performed using the standard protocol (Sigma-Aldrich, St. Louis, MO). Immunostaining was performed using a standard protocol. We incubated sections with primary antibodies to mouse pSmad2/3 (sc-11769, Santa Cruz Biotechnology, 1:100), αVβ6 (BS-5791R, Bioss Antibodies, Woburn, MA, RRID:AB_11042026, 1:100), αVβ3 (BS-1310R, Bioss Antibodies, RRID: AB_10854294, 1:100), αVβ5 (BS-1356R, Bioss Antibodies, RRID: AB_10853044, 1:100), Anti-integrin β8 (orb184308, Biorbyt Inc., 1:100), Anti-Talin (ab71333, Abcam, RRID: AB_2204002, 5 μg/ml), and Aggrecan (AB1031, MilliporeSigma, Burlington, MA, 10 μg/ml) overnight at 4 °C. For immunohistochemical staining, a horseradish peroxidase–streptavidin detection system (Dako, Santa Clara, CA) was subsequently used to detect the immunoactivity followed by counterstaining with hematoxylin (Dako). For immunofluorescent staining, secondary antibodies conjugated with fluorochrome were added, and slides were incubated at room temperature for 1 h while avoiding light. We calculated the ratio of positively stained cells to total chondrocytes in the region of interest in five sequential sections per mouse in each group. For cell signaling, the SV40 cells were grown on glass coverslips collated with 0.1% gelatin. After fixation with 3.7% paraformaldehyde solution, the primary antibody was applied on the coverslip for 1 h at room temperature. After carefully washing, the cells were incubated with secondary antibodies conjugated with fluorochrome for 1 h at room temperature. F-actin was labeled by Phalloidin Conjugates (Sigma-Aldrich, 50 μg/ml) for 40 min at room temperature. For the quantitative analysis of all the immunostaining, we first averaged the positive cell numbers in three randomly selected fields of view for each specimen. The final values shown in the bar chart are the average value of all specimens. The $n$ value represents the sample size (how many specimens) that have been used in the experiments. In the parallel experiment that characterizes the TGFβ activity distribution in the AC, the pSmad2/3 immunofluorescence intensity at the different regions with distinctive mechanical stress was calculated and averaged from three independent specimens. The mechanical stress at different regions was determined by the FE analysis.

**Micro-computed tomography**. We dissected mouse knee joints free of soft tissue, fixed them overnight in 70% ethanol, and analyzed them by high-resolution μCT (Skyscan1172, Bruker, Kontich, Belgium)[79]. We reconstructed and analyzed images using NRecon, v1.6, and CTAn, v1.9, respectively. Three-dimensional model visualization software, CTVol, v2.0, was used to analyze parameters of the

trabecular bone in the metaphysis. The scanner was set at a voltage of 50 kV, a current of 200 μA, and a resolution of 5.7 μm per pixel. Cross-sectional images of the tibial SB were used to perform 3D histomorphometric analysis. We defined the region of interest as covering the whole SB medial compartment, and we used ten consecutive images from the medial tibial plateau for 3D reconstruction and analysis. 3D structural parameters analyzed were TV (total tissue volume, which includes both trabecular and cortical bone), BV/TV, Tb. Th (trabecular thickness), Tb. Sp (trabecular separation), structural model index, Conn. Dn (connectivity density), and Tb. Pf (trabecular pattern factor). The SMI was used to measure rods and plates in the trabecular bone[80]. The trabecular pattern factor (Tb. Pf) was used to reflect the connectedness of trabeculae[81]. The bone surface/volume fraction (BS/BV) was obtained in the 3D pattern by calculating the average ratio of the bone surface area to the volume of mineralized bone in each CT scan sections within the region of interests (bone microarchitecture analysis manual, https://analyzedirect.com/documents/BMA_Manual.pdf).

**Finite element analysis**. Using the Image Processing Language software (IPL, version 5.6, Scanco Medical AG), the tibial plateaus above the growth plate were selected from the μCT images, thresholded, segmented into cartilage and bone, and converted into finite element models by setting each image voxel to an eight-node brick element[82,83]. A uniaxial compression test was simulated by fixing the bottom and prescribing a constant load (10 μN, resulting in average strain of approximately 10% in the cartilage elements) on the top along the loading axis. The applied load was evenly distributed among nodes belonging to cartilage elements at the top surface of the tibial plateau. Bone and cartilage elements were assigned linear elastic material properties: Young's modulus of 15 GPa and 1 MPa and Poisson's ratio of 0.3 and 0.12, respectively. Stresses and strains in the cartilage were calculated using the ABAQUS finite element software (Version 6.11, DASSAULT Systems, Johnston, RI).

**Live cell micromechanical methods**. To detect the cytoskeletal remodeling rate and cell stiffness, we used spontaneous nanoscale tracer motions and magnetic twisting cytometry, respectively. These methods make use of spontaneous and forced motions of RGD-coated ferrimagnetic beads anchored to the cytoskeleton through cell surface integrin receptors, as described in detail elsewhere[84–87]. For these studies, cells were plated on collagen I-coated plastic wells (96-well Remo-vawell, Immulon II, Dynetech). Spontaneous nanoscale movements of individual beads bound to adherent cells were recorded, and bead motions were characterized by computing the mean square displacement of the beads as a function of time [MSD($t$)] (nm²), as previously described[84]. We then created forced bead motions by applying a horizontal magnetic field and twisting with a vertically aligned 20-Gauss magnetic field that varied sinusoidally in time. The resulting oscillatory torque causes beads to rotate along the cell surface[85]: as the beads move, the cell develops internal stress to resist the bead motion. Lateral bead displacements were detected via a charge-coupled device camera (Orca II-ER, Hamamatsu Photonics, Hamamatsu, Japan) attached to an inverted optical microscope (Leica Micro-systems, Bannockburn, IL). We defined the ratio of applied torque to lateral bead displacements as the complex elastic modulus ($g*$) of the cell, $g*(f) = g'(f) + ig''(f)$, where $g'$ is the elastic/storage modulus (cell stiffness), $g''$ is the loss modulus (cell friction), and $i2 = -1$[85]. Cell stiffness was measured over a physiologic range of frequency ($f$) and expressed in units of Pascal per nm (Pa/nm).

**Fourier transform traction microscopy**. Using traction microscopy, we measured the distribution of traction fields arising at the interface between each adherent cell and its substrate. As described in detail elsewhere[86–89], inert elastic polyacrylamide gel blocks were coated with collagen I (0.2 mg/ml) using a photo-activating cross-linker sulfo-SANPAH (Pierce, Rockford, IL). In brief, vehicle or sheared SV40 cells were plated sparsely on the respective gel blocks and allowed to adhere for 2 h. For each adherent cell, images of fluorescent microbeads (0.2 μm in diameter, Molecular Probes, Eugene, OR) embedded near the apical surface of the gel were taken. The fluorescent image of the same region of the gel after cell detachment with trypsin was used as the reference (traction-free) image. The displacement field between a pair of images was then obtained by identifying the coordinates of the peak of the cross-correlation function[88]. From the displacement field and known elastic properties of the gel (Young's modulus of 8000 Pa with a Poisson ratio of 0.48), the traction field was computed using constrained Fourier transform traction cytometry[88]. The computed traction field was used to obtain RMS traction, which is a scalar measure of the cell's contractile strength[88]. RMS traction is expressed in Pascals (Pa).

**TGT assay**. To determine molecular tension on a single integrin–ligand bond exerted by the chondrocyte, we fabricated TGT as described elsewhere[53]. Briefly, a cyclic peptide RGDfK (cRGDfK) ligand was conjugated to the 3′ end of an 18-nucleotide single-stranded DNA (ssDNA; 5-/5Cy3/GGCCCGCAGCGACCACCC/3ThioMC-D/3) by a thiol–maleimide coupling reaction, as well as an N-hydroxysuccinimide ester–amino reaction. A complementary ssDNA (5-GGGTGGTCGCTGCGGGCC-3) with biotin at distinct locations was annealed to the cRGDfK-ssDNA, resulting in dsDNA with different tension tolerances (Ttol, 43 and 54 pN). We immobilized 1 μM of each TGT on a PEGylated glass slide via a neutravidin–biotin interaction. Next, chondrocytes treated with combinations of siRNA (targeting αv or α5 integrin) and shear stress were collected from a culture

dish, resuspended in a serum-free DMEM, and loaded onto the TGT surface at a density of 10⁵ cells/ml. After 30-min or 2-h incubation at 37 °C, the cells were fixed with 4% paraformaldehyde. A differential interference contrast image of the fixed cells and a fluorescence image of the TGT rupture patterns induced by the cells were taken using an epi-fluorescence microscope (Ti-E, Nikon, Tokyo, Japan). The number of adherent cells on each TGT surface and the rupture patterns were further analyzed using the ImageJ software (public domain, version 1.50i).

**Statistics**. Data are presented as mean ± standard error. The comparisons for Osteoarthritis Research Society International scores, bone mass, and micro-architecture among different groups were performed using multifactorial analysis of variance (ANOVA). When ANOVA testing indicated the overall significance of main effects, without interaction between them, the differences between individual time points and sites were assessed by post hoc tests. The level of significance was set at $P < 0.05$. All data analyses were performed using the SPSS, version 15.0, analysis software (SPSS, Chicago, IL).

**Reporting summary**. Further information on research design is available in the Nature Research Reporting Summary linked to this article.

## Data availability

All relevant data that support the findings of this study are available within this published article or available from the corresponding author upon reasonable request. Source data are provided with this paper.

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

## Acknowledgements

This research was supported by National Institute on Aging of the National Institutes of Health under Award Number R01 AG068997 and P01AG066603 (to X.C.). G.Z were partially supported by DOD grant W81XWH-19-1-0222.

## Author contributions

Conceptualization: G.Z. and X.C.; methodology: G.Z., S.Z, B.C.K., and J.H.; investigation: G.Z., Q.G, S.Z., Y.L., C.W., S.Z., R.W., B.C.K., J.H., Y.H., and Y.D.; writing—original draft: G.Z., B.C.K., J.H., and Y.H.; writing—review and editing: G.Z. and X.C.; funding acquisition: X.C. and G.Z.; resources: E.G., T.H., S.A., M.W., and X.C.; supervision: X.E.G., T.H., S.A., and X.C.

## Competing interests

The authors declare no competing interests.
