## [Peer Review File · Nature Communications]

Editorial Note: Parts of this Peer Review File have been redacted as indicated to remove material where no permission to publish could be obtained.

Reviewers' comments:

Reviewer #1 (Remarks to the Author):

In this manuscript, the authors investigated how subchondral bone alterations effect articular cartilage homeostasis. The authors proposed that mechanical stress stimulates talin-centered cytoskeletal reorganization and the consequent increase of cell contractile forces and cell stiffness of chondrocytes, which triggers αV integrin-mediated TGF β activation.

This manuscript is potentially interesting because this is the first report showing that talin has a key role in the mechanosensory system of chondrocytes. However, the present study has the critical concerns. Importantly, the authors did not use chondrocyte-specific talin deficient mice. In addition, although the human sample was available, the authors did not provide the data of talin expression in chondrocytes isolated from human articular cartilage.

Taken together, this manuscript is still immature for publication in Nature Communications in its current form.

Major concerns

1. The importance of talin in the mechanosensory system of chondrocytes cannot be proven without conditional KO mice. The authors should show the phenotypes of chondrocyte-specific talin deficient mice.
2. Because the human sample is available, the authors should provide the data of talin expression in chondrocytes isolated from human articular cartilage.
3. The authors should check the synovial fluid levels of TGF β from the human and the mice in this study.

Reviewer #2 (Remarks to the Author):

In this study, the authors establish a relationship between subchondral bone architecture, mechanical stress in the articular cartilage, integrin activation in chondrocytes, and the disruption of TGF β homeostasis. The study includes an impressively large range of experimental systems and methods. However, essential information about experimental details is often lacking. Specifically, I found not details on what a structural model index or a trabecular pattern factor is, or how the bone surface/volume fraction was measured. The data base for Fig. 1 is unclear (how many cells from how many fields of view from how many bone samples) and should be specified in the figure legend or directly in the figures. Information like „n=10“ is insufficient as it is unclear what „n“ refers to. Scale bars for the μ CT images are missing. Also for many of the other measurements in the following figures, it is unclear how many independent experiments were conducted, how many cells were analyzed ect. The scale bar in Fig. 2a is missing. The color bar for Fig. 2b is difficult to read, and perhaps the authors could consider using the same scale for sham and ACLT operated mice so that the differences become more clear. It is not clear to me how an increased bone roughness results in an increased stress and strain, and how this may affect the average stresses and strains over a larger region. It is also not clear to me how the load of 10 μ N was distributed over the cartilage surface, or how the authors establish that stresses were generally increased in the anterior and interior regions, and generally reduced in the center region. Were the pSmad2/3 immunofluorescence data from the individual samples (n = ?) paired with the corresponding μ CT/FEA data? It is not clear how shear stress was applied to cultured cells. The authors state “120 rpm”, but what item was rotated at 120 rpm and what shear stress that translates to is not specified.

Reviewer #3 (Remarks to the Author):

This is a very interesting paper indicating that activation of TGF-beta is altered in OA cartilage due to changes in mechanical forces in the cartilage due to changes in the subchondral bone. The paper indicates that cartilage stays healthy by the activation of a "normal" level of TGF-beta and that changes outside of this range will lead to disease. The work appears to be carried out reliably although some details are missing in the M & M sections about the used procedures. The paper contributes to our understanding of the enigmatic role of TGF-beta in osteoarthritis.

The authors indicate that change in the subchondral bone leads to changes in TGF-beta activation. The question remains how these changes in the subchondral bone are initiated. This should be discussed.

Immortalized human chondrocytes-SV40 were purchased from Applied Biological Systems. What was the phenotype of the cells used. Do these cells still have a chondrocyte phenotype compared to freshly isolated human chondrocytes. These cells should be characterized better.

A major function of TGF-beta in chondrocytes is inhibition of hypertrophy. Data on synthesis of aggrecan in articular chondrocytes are less clear, in contrast to effects on stem cells, and it is well known that TGF-beta inhibits IGF-I signaling. A factor crucial for aggrecan synthesis by chondrocytes. In my opinion this should be mentioned in the introduction.

"Indeed, superposition of mechanical compression impairs the anabolic effect of TGFβ on chondrocyte matrix production". Is this a well-accepted phenomenon. Is this not a result of the fact that mechanical compression induces TGF-beta synthesis and activation.

It is not well described how human cartilage was categorized. What were the criteria to categorize these samples? What was the size of the cartilage sections examined? How was the normal cartilage determined to be normal?

Typo in line 252.

Line 325. Another explanation could be that normal TGF-beta activation is inhibited and that this leads to the observed histological changes.

Line 33. "In this study, we show that temporal-spatial activation of TGFβ to maintain TGFβ activity within an appropriate range is essential for the maintenance of articular homeostasis and structural integrity;". This is not really demonstrated in this study but has been demonstrated by others. This study shows how aberrant TGF-beta activation might lead to pathology.

Mice were treated with tamoxifen. Not clear how tamoxifen was administered.

In the shear stress experiments. Could the cells in the center be used as controls?

Line 611. "We determined the concentration of active TGF-β1 in the conditioned media by the ELISA". Are these samples activated by acid or heat or not before measurement?

We would like to thank the reviewers for their thoughtful and constructive comments regarding our manuscript. We have addressed all of their concerns and questions brought forth through additional experimentation and clarification. The changes are marked at the left margin in the manuscript.

Reviewer #1 (Remarks to the Author):

In this manuscript, the authors investigated how subchondral bone alterations affect articular cartilage homeostasis. The authors proposed that mechanical stress stimulates talin-centered cytoskeletal reorganization and the consequent increase of cell contractile forces and cell stiffness of chondrocytes, which triggers αV integrin-mediated TGF β activation. This manuscript is potentially interesting because this is the first report showing that talin has a key role in the mechanosensory system of chondrocytes. However, the present study has critical concerns. Importantly, the authors did not use chondrocyte-specific talin deficient mice. In addition, although the human sample was available, the authors did not provide the data of talin expression in chondrocytes isolated from human articular cartilage.

Response: We designed additional experiments to validate the role of talin1 in OA animal models by knocking down the expression of talin1 in chondrocytes through intra-articular injection of talin-1 siRNA. Particularly, immunostaining of talin1 in human articular cartilage to characterize the expression of talin1 in human specimens. Detailed descriptions have been provided below.

Major concerns:

1. The importance of talin in the mechanosensory system of chondrocytes cannot be proven without conditional KO mice. The authors should show the phenotypes of chondrocyte-specific talin deficient mice.

Response: [Redacted] Because this is the only talin-1 flox mouse strain available, we performed intra-articular injection of Talin1 siRNA to knockdown the expression of talin1 in chondrocytes. The *in vivo*

administration of siRNA is a very mature technique and has been tested in several clinical trials^{1,2} (<https://clinicaltrials.gov/ct2/show/NCT01591356>; <https://clinicaltrials.gov/ct2/show/NCT00716014>).

Before the intra-articular injection of talin1 siRNA to the OA animal model, we performed the dose-dependent and time-dependent experiment to determine the optimal dose, frequency, and treatment window. We found that the expression of talin in chondrocytes was significantly inhibited when 10^{-2} nM of siRNA was administered intra-articularly every three days. These results validate that the intra-articular injection of si-RNA successfully mimics the genetic deletion of talin1 in the mouse model with better translational potential

Fig A Immunohistological staining of talin in the articular cartilage of mice tibia plateau with different dose of talin siRNA intra-articular injection. Specimens were harvested at day 3 and day 7 post injection..

(**Fig A**). Our *in vitro* experiments already showed that siRNA talin inhibited of αV integrin–RGD bond under mechanical stress (**Fig 4c** in the manuscript). In the revised manuscript, we further tested the role

of talin in mediating mechanical stress-induced TGF β activation in the OA mouse model. We found that

suppressing the expression of talin by intra-articular injection of talin siRNA significantly downregulated the TGF β signaling in the articular cartilage of mice that subjected to ACLT surgery (Fig. 4i-k in the revised manuscript). This phenomenon was not observed in the sham operated mice. The finding validates that talin is an essential component in the process of α V integrin mediated TGF β activation in articular cartilage particularly under mechanical stress.

2. Because the human sample is available, the authors should provide the data of talin expression in chondrocytes isolated from human articular cartilage.

Response: In the revised manuscript, we performed immunohistochemical staining of Talin1 in the human articular specimens. Our results demonstrated that the talin was expressed by the articular cartilage chondrocytes of both OA-M and OA-S specimens with slightly increased expression in the OA-S specimen (Fig B,

Fig B Immunohistological staining of talin in the articular cartilage of human tibia plateau. OA-M: OA specimen with minimal cartilage degeneration, OA-S: OA specimen with severe cartilage degeneration.

supplemental Fig. 7). The results suggest that possibly both activation and increased expression of talin play a role in α V integrin-mediated TGF β activation during the progression of OA. This finding is consistent and complementary to our in vitro experiments that siRNA talin inhibited its function to enhance the strength of α V integrin–RGD bond under mechanical stress (Fig 5g).

3. The authors should check the synovial fluid levels of TGF β from the human and the mice in this study.

Response: Because of the extremely limited synovial fluid in the mouse knee joint, we were not able to draw synovial fluid from mouse knee joints. We therefore measured the levels of TGF β in the lavage fluid that collected from ACLT mice and sham-operated mice to show TGF β in the synovial fluid by performing ELISA assay. We found that the levels TGF β in lavage fluid of the synovial capsule in ACLT mice were significantly elevated relative to that of sham-operated mice (Fig. C).

Fig C ELISA assay of TGF- β 1 in the lavage fluid of the mice at 1-month post ACLT or sham operation.

Reviewer #2 (Remarks to the Author):

In this study, the authors establish a relationship between subchondral bone architecture, mechanical stress in the articular cartilage, integrin activation in chondrocytes, and the disruption of TGF β homeostasis. The study includes an impressively large range of experimental systems and methods. However, essential information about experimental details is often lacking. Specifically, I found no details on what a structural model index or a trabecular pattern factor is, or how the bone surface/volume fraction was measured.

Response: All of the 3D structural parameters of CT analysis were generated by CTAn software (Bruker, Kontich, Belgium) based on the high-resolution CT images. The structural model index (SMI) is widely used to measure rods and plates in trabecular bone³. It exploits the change in surface curvature that occurs as

a structure varies from spherical (SMI=4) to cylindrical (SMI=3) to planar (SMI=0). It has been shown previously that subchondral trabecular rod loss and plate thickening are essential indicators for the development of osteoarthritis⁴. The trabecular pattern factor (Tb.pf) is a μ CT parameter that has been defined to reflect the connectedness of trabeculae⁵. The basic idea is that the connectedness of structures can be described by the relation of convex to concave surfaces. A lot of concave surfaces represent a well-connected spongy lattice, whereas a lot of convex surfaces indicate a badly connected trabecular lattice. Our previous studies showed that the connectivity of trabecular bone in the subchondral bone of the OA animal model and the human specimen was significantly decreased⁶. The Bone surface/volume fraction (BS/BV) is the ratio of the bone surface area to the volume of mineralized bone (bone microarchitecture analysis manual, https://analyzedirect.com/documents/BMA_Manual.pdf). Elevated BS/BV may indicate less well-connected bony tissues. We therefore used these parameters to reflect the micro-architecture of subchondral bone from different angles. The description of these parameters has been added into the method sections of the revised manuscript.

The database for Fig. 1 is unclear (how many cells from how many fields of view from how many bone samples) and should be specified in the figure legend or directly in the figures. Information like „n=10“ is insufficient as it is unclear what „n“ refers to. Scale bars for the μ CT images are missing. Also for many of the other measurements in the following figures, it is unclear how many independent experiments were conducted, how many cells were analyzed, etc. The scale bar in Fig. 2a is missing.

Response: For the quantitative analysis of all the immunostaining, we firstly averaged the positive cell numbers in 3 randomly selected fields of view for each specimen. The final values shown in the bar chart are the average value of all specimens. The n value represents the sample size (how many specimens) that have been used in the experiments. For the in vitro experiments, the data were generated based on three independent repeats. These descriptions have been added into the figure legends or method section accordingly. The Scale bars for the μ CT images have been added.

The color bar for Fig. 2b is difficult to read, and perhaps the authors could consider using the same scale for sham and ACLT operated mice so that the differences become more clear. It is not clear to me how an increased bone roughness results in increased stress and strain, and how this may affect the average stresses and strains over a larger region. It is also not clear to me how the load of 10 μ N was distributed over the cartilage surface, or how the authors establish that stresses were generally increased in the anterior and interior regions, and generally reduced in the center region.

Response: We have re-constructed Fig 2b and modified color bar with the same scale for sham and ACLT groups accordingly. Increased roughness would result in localized stiffening of the underlying subchondral bone, which would in turn result in stress concentrations in the overlying articular cartilage and uneven distribution of strain, as some areas of the cartilage would experience more stress/strain than before while other will experience lower stress/strain. During the simulations, all nodes on the top surface of the articular cartilage were loaded by an equal amount that summed to be 10 μ N in total transmitted through the joint. As stress/strain plots are displayed on the same color scale, comparing the color between sham and ACLT at the corresponding regions can be representative of the stress/strain values. We chose a 10 μ N as the loading value because it generated an average deformation of about 10% (an average strain of 0.1 along axial direction) of the original cartilage thickness, which is on a similar scale as other papers^{7,8}.

Were the pSmad2/3 immunofluorescence data from the individual samples (n = ?) paired with the corresponding μ CT/FEA data?

Response: The pSmad2/3 immunofluorescence data were generated from three independent specimens. whereas the FE model was established based on the μ CT image of one of the specimens. The description has been added in the method sections of the revised manuscript.

It is not clear how shear stress was applied to cultured cells. The authors state “120 rpm”, but what item was rotated at 120 rpm and what shear stress that translates to is not specified.

Response: The shear stress across the cells on the periphery of the dishes was calculated as $\tau_{\max} = a\sqrt{\rho\eta(2\pi f)^3}$, where a is the radius of orbital rotation (1.75 cm), ρ is the density of the medium (1.0 g/ml), η is the viscosity of the medium (7.5×10^{-3} dynes·s/cm²), and f is the frequency of rotation (rotations/second). Using this equation, shear stress of 6.58 dynes/cm² is achieved at a rotating frequency of 118 rpm. The calculation method has been described in the method section of the manuscript.

Reviewer #3 (Remarks to the Author):

This is a very interesting paper indicating that activation of TGF-beta is altered in OA cartilage due to changes in mechanical forces in the cartilage due to changes in the subchondral bone. The pare indicates that cartilage stays healthy by the activation of a “normal” level of TGF-beta and that changes outside of this range will lead to disease. The work appears to be carried out reliable although some details are missing in the M & M sections about the used procedures. The paper contributes to our understanding of the enigmatic role of TGF-beta in osteoarthritis.

The authors indicate that change in the subchondral bone leads to changes in TGF-beta activation. The question remains of how these changes in the subchondral bone are initiated. This should be discussed.

Response: In the previous study⁶, we found that osteoclastic bone resorption was significantly elevated in the subchondral bone at the early stage of osteoarthritis. As a result, excessive active TGF-beta was liberated and accumulated in the bone marrow cavity. The mesenchymal stem cells and the osteoprogenitors clustered in the bone marrow cavity and resulted in aberrant bone formation because they can't be recruited to the bone resorption site following the normal TGF-beta gradient. We further validated our findings by conditionally knocking out the TGF-beta type II receptor in the mesenchymal stem cells. We found that the aberrant bone formation and structural changes in the subchondral bone and the degeneration of articular cartilage were significantly attenuated in the OA mouse models. These findings were published in *Nature Medicine* (2013) and have been discussed in the present manuscript.

Immortalized human chondrocytes-SV40 were purchased from Applied Biological Systems. What was the phenotype of the cells used? Do these cells still have a chondrocyte phenotype compared to freshly isolated human chondrocytes? These cells should be characterized better.

Response: Thanks for the suggestion. We further characterized the expression of chondrocyte markers (Aggrecan, type II collagen, Sox9) in the SV40 chondrocyte cell line using immunofluorescent staining, and Q-PCR. The primary isolated mouse chondrocytes were used as positive control while Raw264.7 cell line as a negative control. We found that the expression of the chondrocyte markers in SV40 was comparable to that of the primary chondrocytes. The results have been added to the revised manuscript as the new supplementary figure 4.

A major function of TGF-beta in chondrocytes is inhibition of hypertrophy. Data on the synthesis of aggrecan in articular chondrocytes are less clear, in contrast to effects on stem cells, and it is well known

that T^αG^β-beta inhibits IGF-I signaling. A factor crucial for aggrecan synthesis by chondrocytes. In my opinion this should be mentioned in the introduction.

Response: Thanks for the suggestion. This concept has been added to the introduction section.

“Indeed, superposition of mechanical compression impairs the anabolic effect of TGF β on chondrocyte matrix production”. Is this a well-accepted phenomenon. Is this not a result of the fact that mechanical compression induces TGF-beta synthesis and activation.

Response: This statement was brought up based on two reports in the literature. In the *ex vivo* study, Dr. Levenston et.al. found that TGF β -1 significantly stimulates the matrix protein production of the bovine cartilage explants. However, the superposition of static mechanical compression inhibited the matrix production in the presence of TGF β ⁹. In the *in vitro* study, Dr. Clark T. Hung and his team found that dynamic deformational loading applied concurrently with TGF- β 3 supplementation yielded significantly lower overall mechanical properties and matrix protein synthesis (glycosaminoglycan and type II collagen) in 3D cultured bovine chondrocytes¹⁰. In the present study, we found that abnormal mechanical loading induces excessive TGF β activation in the articular cartilage. We used these references to support our point of view that abnormal loading may also exacerbate the detrimental effect of high levels of TGF β on cartilage homeostasis.

It is not well described how human cartilage was categorized. What were the criteria to categorize these samples? What was the size of the cartilage sections examined? How was the normal cartilage determined to be normal?

Response: The OA specimens were collected from the tibial plateau specimens of OA patients that received total knee replacement. The specimens were examined by raw eyes and one OA-M and one OA-S sample at 1*1*1 cm will be collected from each OA specimen. The samples with intact cartilage coverage under the raw eye were categorized into the OA-M group while samples with severe cartilage damage were categorized into the OA-S group. The healthy knee specimens were purchased from the National Disease Research Interchange (Philadelphia, PA) to serve as controls. These knee joint specimens are collected from human donors without a history of OA. This information has been added to the method section.

Typo in line 252.

Response: Thanks for pointing out the typo. The typo has been corrected.

Line 325. Another explanation could be that normal TGF-beta activation is inhibited and that this leads to the observed histological changes.

Response: Yes. We agree with the reviewer that this possibility can not be completely excluded. Some of the other RGD binding integrins such as α 5 β 1 has been reported to be involved in the GAG and proteoglycan synthesis of chondrocytes¹¹. We thus interpreted that non-specific neutralization of other RGD binding integrins may contribute to the phenomenon (mild proteoglycan loss was also observed in RGD-treated sham-operated mice). This discussion has been added to the revised manuscript.

Line 33. “In this study, we show that temporal-spatial activation of 334 TGF β to maintain TGF β activity within an appropriate range is essential for the maintenance of 335 AC metabolic homeostasis and

structural integrity;”. This is not really demonstrated in this study but has been demonstrated by others. This study shows how aberrant TGF-beta activation might lead to pathology.

Response: Thanks for pointing out this. This illustration has been revised accordingly.

Mice were treated with tamoxifen. Not clear how tamoxifen was administered.

Response: One week before surgery, we treated each group with 80 mg/kg bodyweight of tamoxifen twice a week for 37 days and euthanized the mice at 30 days postoperatively. The dosage and treatment frequency were determined based on our preliminary experiment. This information has been added to the method section.

In the shear stress experiments. Could the cells in the center be used as controls?

Response: In the shear stress experiments, we didn’t use the cells in the center as controls because these cells still subjected to low levels of shear stress.

Line 611. “We determined the concentration of active TGF-β1 in the 612 conditioned media by the ELISA”. Are these samples activated by acid or heat or not before measurement?

Response: We measured the levels of both active and total TGFβ using ELISA assay. As instructed by the vendor-provided manual, the active TGFβ was measured directly from the conditioned media while the total TGFβ was measured after treating with acid. This description has been added to the method section.

- 1 Saw, P. E. & Song, E. W. siRNA therapeutics: a clinical reality. *Sci China Life Sci*, doi:10.1007/s11427-018-9438-y (2019).
- 2 Chakraborty, C., Sharma, A. R., Sharma, G., Doss, C. G. P. & Lee, S. S. Therapeutic miRNA and siRNA: Moving from Bench to Clinic as Next Generation Medicine. *Mol Ther Nucleic Acids* **8**, 132-143, doi:10.1016/j.omtn.2017.06.005 (2017).
- 3 Hildebrand, T. & Ruegsegger, P. Quantification of Bone Microarchitecture with the Structure Model Index. *Comput Methods Biomech Biomed Engin* **1**, 15-23, doi:10.1080/01495739708936692 (1997).
- 4 Chen, Y. *et al.* Subchondral Trabecular Rod Loss and Plate Thickening in the Development of Osteoarthritis. *J Bone Miner Res* **33**, 316-327, doi:10.1002/jbmr.3313 (2018).
- 5 Hahn, M., Vogel, M., Pompesius-Kempa, M. & Dellling, G. Trabecular bone pattern factor--a new parameter for simple quantification of bone microarchitecture. *Bone* **13**, 327-330, doi:10.1016/8756-3282(92)90078-b (1992).
- 6 Zhen, G. *et al.* Inhibition of TGF-beta signaling in mesenchymal stem cells of subchondral bone attenuates osteoarthritis. *Nat Med* **19**, 704-712, doi:10.1038/nm.3143 (2013).
- 7 Abusara, Z., Von Kossel, M. & Herzog, W. In Vivo Dynamic Deformation of Articular Cartilage in Intact Joints Loaded by Controlled Muscular Contractions. *PLoS One* **11**, e0147547, doi:10.1371/journal.pone.0147547 (2016).
- 8 Cao, L., Youn, I., Guilak, F. & Setton, L. A. Compressive properties of mouse articular cartilage determined in a novel micro-indentation test method and biphasic finite element model. *J Biomech Eng* **128**, 766-771, doi:10.1115/1.2246237 (2006).
- 9 Imler, S. M., Doshi, A. N. & Levenston, M. E. Combined effects of growth factors and static mechanical compression on meniscus explant biosynthesis. *Osteoarthritis Cartilage* **12**, 736-744, doi:10.1016/j.joca.2004.05.007 (2004).

- 10 Lima, E. G. *et al.* The beneficial effect of delayed compressive loading on tissue-engineered cartilage constructs cultured with TGF-beta3. *Osteoarthritis Cartilage* **15**, 1025-1033, doi:10.1016/j.joca.2007.03.008 (2007).
- 11 Pulai, J. I., Del Carlo, M., Jr. & Loeser, R. F. The alpha5beta1 integrin provides matrix survival signals for normal and osteoarthritic human articular chondrocytes in vitro. *Arthritis Rheum* **46**, 1528-1535, doi:10.1002/art.10334 (2002).

REVIEWER COMMENTS

Reviewer #1 (Remarks to the Author):

The authors have adequately addressed my concerns and as a consequence the findings have been significantly strengthened and extended.

Reviewer #2 (Remarks to the Author):

Several of my questions relating to the methods have not been fully answered by the authors. In particular

- How was the bone surface/volume fraction measured? Obviously, I know what a ratio or fraction is, but I don't know how the surface area or the bone volume was measured. From a single slice, based on some stereological considerations? Or from a 3-D volume? Based on which image segmentation approach? The referenced software manual does not answer these questions.

- It is not clear to me how structural changes in the SB cause increased stress and strain in the AC, which is a (if not The) central tenet of the study. I understand that this comes out of the FE analysis, but I lack an intuitive understanding. The authors state that the interface between SB and AC has a rugged appearance in OA mice, which I can well imagine causes locally increased stress at the AC. But the ruggedness of the interface has not been characterized in this study, whereas the structural changes in the SB have been extensively characterized, but they (in my opinion) cannot be the main reason for the increased stress in the AC. If they are, please explain. I am sorry but I found the explanation given in the rebuttal letter not helpful. In this connection, the authors state that "all nodes on the top surface of the articular cartilage were loaded by an equal amount". How equidistant were the nodes on the top surface, and was the top surface in the simulations flat or curved?

- It is not clear how shear stress was applied to cultured cells. The authors state "120 rpm", but what item was rotated at 120 rpm? Rotated around what?

Reviewer #3 (Remarks to the Author):

The authors addressed all concerns with this manuscript

We would like to thank the reviewers for the positive feedback on our first-round revision. We have modified the manuscript and add details to further clarify the method brought forth by Reviewer 2. The changes are highlighted in the revised manuscript.

Reviewer #1 (Remarks to the Author):

The authors have adequately addressed my concerns and as a consequence the findings have been significantly strengthened and extended.

Thanks for the encouraging and constructive comments from the reviewer.

Reviewer #2 (Remarks to the Author):

Several of my questions relating to the methods have not been fully answered by the authors. In particular

- How was the bone surface/volume fraction measured? Obviously, I know what a ratio or fraction is, but I don't know how the surface area or the bone volume was measured. From a single slice, based on some stereological considerations? Or from a 3-D volume? Based on which image segmentation approach? The referenced software manual does not answer these questions.

Response: The bone surface/volume (BS/BV) is a parameter of the micro-CT scan, which represents the ratio of the segmented bone surface to the segmented bone volume¹. It is conventionally computed by triangulation of the trabecular surface using a marching cubes algorithm. The BS/BV was obtained in the 3D pattern by calculating the average of BS/BV in each CT scan sections within the region of interests. The description of BS/BV has been modified in the method section of the revised manuscript.

- It is not clear to me how structural changes in the SB cause increased stress and strain in the AC, which is a (if not The) central tenet of the study. I understand that this comes out of the FE analysis, but I lack an intuitive understanding. The authors state that the interface between SB and AC has a rugged appearance in OA mice, which I can well imagine causes locally increased stress at the AC. But the ruggedness of the interface has not been characterized in this study, whereas the structural changes in the SB have been extensively characterized, but they (in my opinion) cannot be the main reason for the increased stress in the AC. If they are, please explain. I am sorry but I found the explanation given in the rebuttal letter not helpful. In this connection, the authors state that "all nodes on the top surface of the articular cartilage were loaded by an equal amount". How equidistant were the nodes on the top surface, and was the top surface in the simulations flat or curved?

Response: The effect of structural changes in the subchondral bone on stress/strain distributions in the articular cartilage has been demonstrated and explored in detail in previous studies². It was shown that by creating local stiffening (densification) in the subchondral bone that mimics an osteoarthritic bone phenotype, stresses in the overlaying cartilage could be elevated by up to 50%. The explanation for this observation was that sclerotic subchondral bone has reduced ability to attenuate and distribute the load evenly throughout the joint, which subsequently increases stress and creates uneven stress distributions in the overlying articular cartilage, as the cartilage and bone work together in load-bearing in the joint.

In the present study, local increased BV/TV and reduced SMI (indicating a more plate-like structure) are signs that the subchondral bone has become sclerotic, which is stiffer and more unevenly distributed. The surface ruggedness is a manifestation of this change. Changes in cartilage stress and strain are not entirely

due to the surface differences but are due to changes in the entirety of the subchondral bone. It is therefore consistent with prior findings that cartilage stress can be elevated due to changes in the subchondral bone. We have edited the FEA results section and corresponding discussion to address this important point.

In creating the FEA models in this analysis, each voxel (5.7 μm along each side) from the μCT image was converted into an 8-node cubic element. In the resulting mesh, each cartilage and bone element is therefore a cubic element with 5.7 μm side length. Since nodes are the 8 corners of each element, neighboring nodes are 5.7 μm apart. The loaded node-set, “all nodes on the top surface of the articular cartilage”, are nodes located on the top surface of cartilage elements that are located at the top of the cartilage volume. The loaded node-set, therefore, forms a shape that follows the natural geometry of the cartilage surface and is therefore curved. We have added more details in the FEA methods section to clarify the ambiguities.

- It is not clear how shear stress was applied to cultured cells. The authors state “120 rpm”, but what item was rotated at 120 rpm? Rotated around what?

Response: We cultured the cells in a 6 well plate that contains in Dulbecco’s Modified Eagle Medium (DMEM) with serum-free media supplement (Nutridoma-SP, Roche, Basel, Switzerland). The orbital shear stress was applied at 37°C in the presence of CO₂ by placing the 6-well plates on an orbital shaker (Thermo multipurpose rotator, ThermoFisher Scientific, Waltham, MA). The rotation of the culture media, therefore, generated shear stress on the cells. The shear stress across the cells on the periphery of the dishes was calculated as $\tau_{\text{max}}=a\sqrt{\rho\eta}(2\pi f)^3$, where a is the radius of orbital rotation (1.75 cm), ρ is the density of the medium (1.0 g/ml), η is the viscosity of the medium (7.5×10^{-3} dynes·s/cm²), and f is the frequency of rotation (rotations/second). Using this equation, shear stress of 6.58 dynes/cm² is achieved at a rotating frequency of ~ 120rpm. The methods section was modified to clarify the ambiguities.

Reviewer #3 (Remarks to the Author):

The authors addressed all concerns with this manuscript

Thanks for the efforts and time of the reviewer.

References:

- 1 Bouxsein, M. L. *et al.* Guidelines for assessment of bone microstructure in rodents using micro-computed tomography. *J Bone Miner Res* **25**, 1468-1486, doi:10.1002/jbmr.141 (2010).
- 2 Radin, E. L. *et al.* Response of joints to impact loading. 3. Relationship between trabecular microfractures and cartilage degeneration. *J Biomech* **6**, 51-57, doi:10.1016/0021-9290(73)90037-7 (1973).

REVIEWERS' COMMENTS

Reviewer #2 (Remarks to the Author):

The authors express shear stress at some place in units of rpm, which is incorrect. The formula given for estimating the maximum shear stress does not seem to give the correct units for fluid shear stress. Where is this formula coming from?

We would like to thank reviewer 2 for his/her time and effort in reviewing our manuscript again. The following is our response to the question raised by reviewer 2.

Reviewer #2 (Remarks to the Author):

The authors express shear stress at some place in units of rpm, which is incorrect. The formula given for estimating the maximum shear stress does not seem to give the correct units for fluid shear stress. Where is this formula coming from?

Response: The “rpm” has been changed to “dynes/cm²” for the description of shear stress in the main text and figure legends. The equation for calculating the shear stress has been modified so that the square root sign applies to the rest of the equation, not just to density.